# Engineering paralog-specific PSD-95 recombinant binders as minimally interfering multimodal probes for advanced imaging techniques

**Charlotte Rimbault[1†], Christelle Breillat[1†], Benjamin Compans[1†], Estelle Toulmé[1†], Filipe Nunes Vicente[1], Monica Fernandez-Monreal[2], Patrice Mascalchi[2], Camille Genuer[1], Virginia Puente-Muñoz[1], Isabel Gauthereau[1], Eric Hosy[1], Stéphane Claverol[3], Gregory Giannone[1], Ingrid Chamma[1], Cameron D Mackereth[4], Christel Poujol[2], Daniel Choquet[1], Matthieu Sainlos[1]\***

[1]University of Bordeaux, CNRS, Interdisciplinary Institute for Neuroscience, IINS, UMR 5297, Bordeaux, France; [2]University of Bordeaux, CNRS, INSERM, Bordeaux Imaging Center, BIC, UMS 3420, US 4, Bordeaux, France; [3]University of Bordeaux, Bordeaux Proteome, Bordeaux, France; [4]University of Bordeaux, Inserm U1212, CNRS UMR 5320, IECB, Pessac, France

**\*For correspondence:**
sainlos@u-bordeaux.fr

[†]These authors contributed equally to this work

**Competing interest:** The authors declare that no competing interests exist.

**Abstract** Despite the constant advances in fluorescence imaging techniques, monitoring endogenous proteins still constitutes a major challenge in particular when considering dynamics studies or super-resolution imaging. We have recently evolved specific protein-based binders for PSD-95, the main postsynaptic scaffold proteins at excitatory synapses. Since the synthetic recombinant binders recognize epitopes not directly involved in the target protein activity, we consider them here as tools to develop endogenous PSD-95 imaging probes. After confirming their lack of impact on PSD-95 function, we validated their use as intrabody fluorescent probes. We further engineered the probes and demonstrated their usefulness in different super-resolution imaging modalities (STED, PALM, and DNA-PAINT) in both live and fixed neurons. Finally, we exploited the binders to enrich at the synapse genetically encoded calcium reporters. Overall, we demonstrate that these evolved binders constitute a robust and efficient platform to selectively target and monitor endogenous PSD-95 using various fluorescence imaging techniques.

## Editor's evaluation

This is a valuable manuscript that develops binders for visualizing postsynaptic protein PSD95 endogenously using a variety of microscopy approaches. Compelling evidence is provided for validating the use of these new imaging probes. These probes should prove useful for visualizing the post-synaptic density in both live and fixed neuronal cells using live cell imaging as well as super-resolution microscopy.

## Introduction

Fluorescence microscopy constitutes nowadays an essential method for cell biologists to monitor the localization and function of most proteins. The discovery of the green fluorescent protein (GFP) and its application as a gene-fused reporter, together with the progress that followed with the isolation and evolution of variants that span the close-UV to near-IR spectrum with various photo-physical and

-chemical properties, have largely contributed to the wide spreading of this approach (*Rodriguez et al., 2017*). Alternative labeling methods such as those relying on engineered enzymes have further expanded the possibilities of imaging approaches by allowing the direct coupling of high-performance organic dyes (*Lavis, 2017*; *Xue et al., 2015*). In parallel, technical breakthroughs in imaging methods have allowed to overcome the diffraction limit and are now enabling optical imaging of biological samples at the nanoscale (*Liu et al., 2015*; *Sahl et al., 2017*; *Schermelleh et al., 2019*). However, while these advances have expanded the scope of application of fluorescence imaging techniques, they have also generated a pressing need for improved labeling strategies (*Choquet et al., 2021*).

Indeed, the capacity to accurately investigate by fluorescence imaging the dynamics of endogenous proteins still constitutes a technical challenge. Antibodies, when available, can only be used against ectodomain-presenting proteins and still suffer from their large size and divalency. In parallel, the main drawbacks of most alternative labeling strategies for proteins (fluorescent protein, enzyme, or tag genetic fusions) are associated with non-physiological regulation of the modified gene expression level and the potential impact of the fusion on the protein of interest function. Recent developments in gene-editing methods (*Bukhari and Müller, 2019*) provide efficient means to circumvent the issue of expression level by directly modifying the endogenous gene, but their implementation is still not straightforward and furthermore intrinsically involves modifying the target protein with a fluorescent tag that can alter its function.

In this context, with the recent progress in directed evolution techniques, recombinant small domain binders capable of specifically recognizing endogenous proteins without impairing their function constitute a promising avenue for the development of minimally invasive and interfering labeling probes (*Bedford et al., 2017*; *Dong et al., 2019*; *Helma et al., 2015*). The increasing diversity in terms of validated molecular scaffolds, such as antibody fragments (scFv or V$_H$H) (*Muyldermans, 2021*) or other domains (DARPins [*Boersma and Plückthun, 2011*], monobodies [*Sha et al., 2017*], affimers [*Tiede et al., 2017*], etc.), provides a large variety of randomized surfaces that can recognize and bind virtually any protein of interest. In addition to their recombinant nature, which facilitates their characterization and allows further engineering – notably to convert them into fluorescent probes – these tools importantly alleviate the need to directly alter the gene of interest. Additionally, their small size allows to bring fluorophores coupled to the engineered evolved domain in close proximity of the targeted protein for advanced imaging techniques.

Two recent studies (*Fukata et al., 2013*; *Gross et al., 2013*) have applied such a strategy to PSD-95, the major postsynaptic scaffold protein at excitatory synapses (*Chen et al., 2005*; *Cheng et al., 2006*), by evolving recombinant binders as key recognition modules for developing imaging probes. PSD-95 plays a key role in organizing receptors, ion channels, adhesion proteins, enzymes, and cytoskeletal proteins at excitatory synapses (*Won et al., 2017*; *Zhu et al., 2016*). As a consequence, up- or downregulation of PSD-95 results in critical alterations in synapse morphology and function (*Won et al., 2017*). In particular, overexpression of fluorescent protein-fused PSD-95 for imaging purposes is phenotypically marked and leads to an increase in dendritic spine number and size, as well as frequency and amplitude of miniature excitatory postsynaptic currents (mEPSCs) and affects synaptic plasticity (*El-Husseini et al., 2000*; *Nikonenko et al., 2008*; *Zhang and Lisman, 2012*). PSD-95 constitutes therefore an ideal candidate for developing labeling strategies that do not affect the protein expression levels. By exploiting evolved binders, a single-chain variable fragment (PF11) (*Fukata et al., 2013*) and a $^{10}$FN3-derived domain/monobody (PSD95.FingR) (*Gross et al., 2013*), the two groups have been able to directly label endogenous PSD-95. However, in both cases, the precise epitopes remain non-characterized, and, furthermore, one of the binders, PSD95.FingR, can also recognize SAP97 and SAP102, two closely related proteins (*Gross et al., 2013*; *Li et al., 2018*). The latter point may constitute a clear limitation, and, additionally, the lack of defined epitopes questions the possibility of PSD-95 function perturbation.

Using a phage display selection approach with a $^{10}$FN3-derived library, we have recently isolated and characterized three monobodies targeting PSD-95 (*Rimbault et al., 2019*). The clones were targeted against PSD-95 tandem PDZ domains and showed remarkable specificity for PSD-95, in particular when considering the high-sequence conservation of paralogs (SAP97, SAP102, and PSD-93). Importantly, all the clones recognized epitopes situated outside of the PDZ domain-binding groove in regions not subjected to post-translational modifications. These properties represent a prerequisite to identify binders having a minimal impact on the tandem domain function and consequently on the

full-length protein. As such, they constitute ideal candidates to engineer and develop minimally interfering imaging probes to monitor endogenous PSD-95.

We describe here the exploitation of specific PSD-95 binders as a platform to develop a series of labeling tools for the endogenous synaptic scaffold protein as well as excitatory synapses targeting modules. We first evaluated the potential impact of each evolved [10]FN3 domain binding on PSD-95 function as well as their capacity to be exploited as intrabody-type imaging tools. The selected binders were further engineered to allow their use in various super-resolution imaging (SRI) modalities (stimulated emission depletion [STED], photoactivation localization microscopy [PALM], and DNA-PAINT). Finally, beyond their direct exploitation as PSD-95 reporters, we validated the strategy to use their binding properties to enrich and address protein-based sensors to the postsynaptic density with the genetically encoded calcium reporter GCaMP6/7f (*Chen et al., 2013*; *Dana et al., 2019*). We termed the approach ReMoRA (Recombinant binding Modules for minimally interfering Recognition and Addressing of endogenous protein targets) as a sub-class of the intrabody general use with applications in fluorescence imaging where emphasis is set on the absence of interference with the targeted protein function.

## Results

### Impact of Xph15/18/20 on PSD-95 PDZ domains function

We have recently selected and isolated [10]FN3-derived clones that bind to the tandem PDZ domains of PSD-95 (*Rimbault et al., 2019*; *Figure 1a*). Three of the evolved [10]FN3 domains, which displayed specific recognition of the target, were characterized in depth in particular with respect to the identification of their respective epitope. Two monobodies (Xph15 and Xph20) shared a similar epitope situated on PDZ domain 1 on the opposite side of the surface compared to the canonical functional region of the domain. Indeed, as protein–protein interaction modules, the principal function of PDZ domains is to bind the C-terminus of their protein partner via a defined solvent-exposed groove. The third monobody (Xph18) presented an extended epitope that spread on both domains 1 and 2, also in regions distant from the two binding grooves. As the three evolved binders did not directly block the two PDZ domain-binding sites, we envisaged their use as minimally interfering targeting modules.

In an initial step prior to designing tools that target endogenous PSD-95, we sought to further characterize the binding properties of the three monobodies in the context of the tandem PDZ domains function. We first used in-solution NMR to evaluate whether the PDZ domain-binding properties to cognate ligands were affected by the presence of either of the clones (*Figure 1b*, *Figure 1—figure supplement 1*). A peptide derived from the C-terminus of a known PSD-95 PDZ domain binder, the auxiliary AMPA receptor (AMPAR) subunit stargazin (Stg), was titrated against a [15]N-labeled PSD-95 tandem construct containing PDZ domains 1 and 2. In addition, the peptide was titrated against the same [15]N-labeled PSD-95 construct pre-bound with either Xph15, Xph18, or Xph20. A series of 2D [15]N-HSQC spectra were used to follow PSD-95 residues during each titration, and in all cases residues on both PDZ1 and PDZ2 were able to fully interact with the Stg peptide. Qualitatively, each of the reporter residues shown in *Figure 1b* has similar titration behavior and final crosspeak positions in the [15]N-HSQC spectra, and therefore supports the fact that the peptide binding is generally unaffected by the presence of the binders. Conversely, by looking at residues at the PSD-95 and binder interface, the added Stg peptides also did not detectably affect the binding of the Xph monobodies (*Figure 1—figure supplement 1*). These results confirm the simultaneous binding of both PDZ domain ligand and monobody and indicate that the PDZ domain-binding properties are not detectably modified in the presence of the evolved Xph binders.

In parallel, we also set up fluorescence polarization assays to determine the binding affinity of representative PSD-95 PDZ domain ligands in the presence or absence of the monobodies. To this end, we used FITC-labeled peptides derived from the last 15 amino acids of stargazin as probes and the recombinant tandem PDZ domains 1 and 2. In order to minimize the effects of varying concentrations of the PDZ domains and the clones, we performed competition assays at constant concentrations of the monobodies, the PDZ domains, and the reporter probe. The potential effect of the evolved [10]FN3 domain binding was first assessed using a divalent ligand titrated with a non-fluorescent divalent competitor both derived from stargazin as a model for complex multivalent interactions (*Figure 1c*, *Figure 1—figure supplement 2*; *Sainlos et al., 2011*). Competitions performed in the absence of

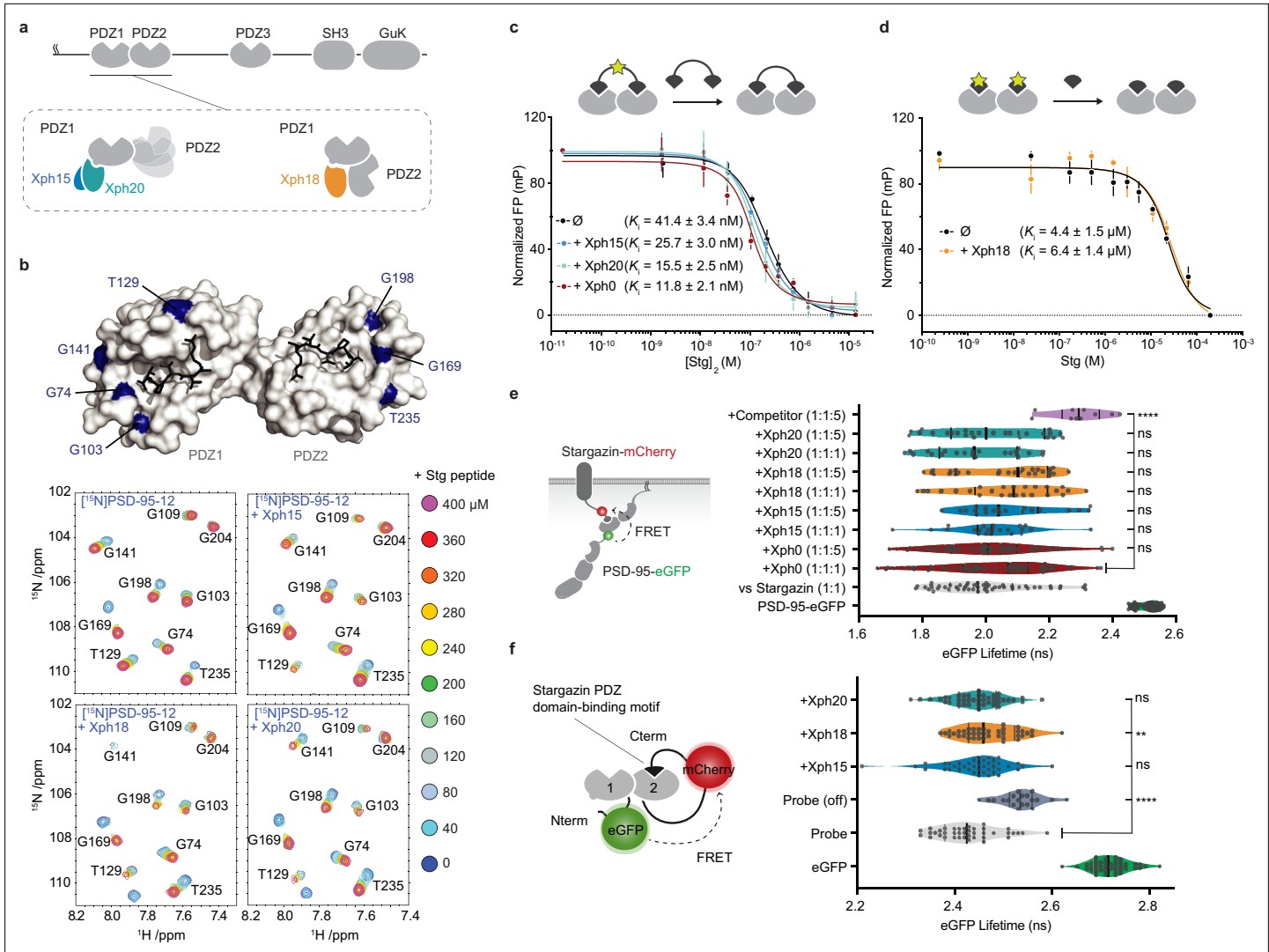

**Figure 1.** Evaluation of the impact of evolved $^{10}$FN3 domains binding on the PDZ domains function. (**a**) PSD-95 domain organization and binding models of the three clones investigated. (**b**) Titrations of a monovalent stargazin-derived peptide against PSD-95-12 in the absence or presence of Xph15, Xph18, and Xph20. Surface representations of PSD-95 tandem PDZ domains (PDB ID 3GSL, domain 1 on the left and domain 2 on the right) with ligand modeled in (RTTPV derived from stargazin C-terminus and aligned from PDB ID 3JXT, black sticks) and with location of the residues annotated in the NMR titration spectra in blue: Gly74, Gly103, Thr129, and Gly141 report on stargazin binding to PDZ1; Gly169, Gly198, and Thr235 report on stargazin binding to PDZ2. Selected region of an overlay of $^1$H,$^{15}$N-HSQC spectra corresponding to 200 µM of [$^{15}$N]PSD-95-12 titrated with 0, 40, 80, 120, 160, 200, 240, 280, 320, 360, and 400 µM peptide ligand based on the C-terminus of stargazin (Stg) in the absence of evolved binder or in complex with 240 µM of Xph15, Xph18, or Xph20. Complete spectra can be found in *Figure 1—figure supplement 1*. (**c**) Competitive fluorescence polarization titrations between divalent stargazin-derived ligands and PSD-95-12 with or without Xph clones (5 µM each, mean ± SD of three independent titrations). (**d**) Competitive fluorescence polarization titrations between monovalent stargazin-derived ligands and PSD-95-12 with or without Xph18 (20 µM, mean ± SD of three independent titrations). (**e**) Lifetime of eGFP inserted in PSD-95 in the presence of stargazin (acceptor-containing protein) and indicated constructs (molar ratio of DNA constructs specified as donor:acceptor:ligand). Violin plots show median, first and third quartile, and all individual data points (each corresponding to a single cell) pooled from at least two independent experiments. Statistical significance determined by one-way ANOVA followed by Dunnett's multiple-comparison test. (**f**) Lifetime of eGFP in a PSD-95-12-derived FRET reporter system in the presence of indicated constructs (used at five molar equivalents of DNA compared to the FRET probe). Violin plots show median, first and third quartile, and all individual data points (each corresponding to a single cell) pooled from at least two independent experiments. Statistical significance determined by one-way ANOVA followed by Dunnett's multiple-comparison test.

The online version of this article includes the following source data and figure supplement(s) for figure 1:

**Source data 1.** Spreadsheet with the normalized fluorescence polarization data (*Figure 1c*).

**Source data 2.** Spreadsheet with the normalized fluorescence polarization data (*Figure 1d*).

**Source data 3.** Spreadsheet with the raw fluorescence lifetime data (*Figure 1e*).

*Figure 1 continued on next page*

*Figure 1 continued*

**Source data 4.** Spreadsheet with the raw fluorescence lifetime data (*Figure 1f*).

**Figure supplement 1.** [$^{15}$N]-HSQC spectra collected on 200 µM [$^{15}$N]PSD-95-12 and titrated in with increasing concentrations of a monovalent stargazin-derived peptide (Stg) (**a**) in the absence of binder or in the presence of 240 µM Xph15 (**b**), Xph18 (**c**), or Xph20 (**d**).

**Figure supplement 2.** Fluorescence polarization titrations.

**Figure supplement 2—source data 1.** Spreadsheet with the normalized fluorescence polarization data (*Figure 1—figure supplement 2a*).

**Figure supplement 2—source data 2.** Spreadsheet with the normalized fluorescence polarization data (*Figure 1—figure supplement 2b*).

**Figure supplement 2—source data 3.** Spreadsheet with the normalized fluorescence polarization data (*Figure 1—figure supplement 2c*).

**Figure supplement 2—source data 4.** Spreadsheet with the normalized fluorescence polarization data (*Figure 1—figure supplement 2d*).

**Figure supplement 3.** Expression regulation system.

ligand or with a naïve clone (Xph0 that does not bind PSD-95) (*Rimbault et al., 2019*) were similar to the ones obtained with Xph15 and 20. In contrast, the presence of Xph18 impaired binding of the fluorescent divalent probe. This effect was abolished, in agreement with the NMR observations, by the use of monovalent stargazin-derived probe and competitor (*Figure 1d*, *Figure 1—figure supplement 2*). These results suggest that the observed inhibitory effect results from the conformational constraints imposed on the two domains orientation by Xph18 binding rather than from the blocking or direct impairment of the PDZ domain-binding ability.

Together, the NMR study and the fluorescence polarization assay indicate that the binding of, on the one hand, Xph clones and, on the other hand, PDZ domain ligands are independent events that are not detectably affected by long-range conformational modifications. However, we note that due to the constraints imposed by Xph18 binding on the conformational flexibility of the two PDZ domains, certain complex interactions may be impaired.

Next, we investigated whether these properties were preserved in complex cellular environments. We therefore evaluated by a FRET/FLIM assay in cell lines the binding of PSD-95 to its partners (represented here by stargazin) via its PDZ domains in the presence and absence of the monobodies. We used both the recombinant full-length proteins (*Figure 1e*) as well as a reporter system that focused on interactions mediated by PDZ domains 1 and 2 (*Figure 1f*). In both cases, even at high molar ratio, we could not detect any significant effect on the measured donor lifetime associated with the binding of either Xph15, Xph18, or Xph20. For the full-length PSD-95 system, the median lifetimes obtained in the presence of the three clones, even at a fivefold molar ratio in the transfected plasmids, were within the variability observed in the presence of a naïve clone (Xph0, between 2.0 and 2.1 ns). On the contrary, co-transfection of a soluble PDZ domain (PSD-95 second PDZ domain, termed here competitor) clearly increased the lifetime to 2.3 ns. Results with the FRET probe based on PDZ domains 1 and 2 were comparable, with an absence of significant modification of the probe lifetime in the presence of the monobodies in comparison to a mutant of the probe in which the PDZ domain-binding motif was deleted (Probe off). A moderate effect was observed by statistical analysis in the case of Xph18, which could be attributed to the constraint imposed by its binding to both PDZ domains 1 and 2. In agreement with the NMR and fluorescence polarization experiments, these results therefore indicate that the primary function of PSD-95 PDZ domains as protein–protein interaction modules is not detectably affected by the interaction with any of the three recombinant binders in a model cellular environment.

## Impact of Xph15/18/20 on PSD-95 function

The main function of PSD-95 is to organize transmembrane receptors such as glutamate receptors at the postsynaptic density and link them to intracellular signaling molecules. Among these, the PSD-95 PDZ domain-mediated interactions with AMPARs through the TARP auxiliary subunits have been particularly well characterized. We and others have previously shown that impairment of the interactions by genetic (*Bats et al., 2007*) or chemical means (*Sainlos et al., 2011*) resulted in a reduction of AMPAR synaptic currents and increased lateral mobility.

In order to rule out any possible effect of the monobodies on endogenous PSD-95 properties, we first evaluated in hippocampal neuron primary cultures whether the presence and binding of the Xph monobodies could impact AMPAR organization and function. To this end, and anticipating exploitation of the evolved $^{10}$FN3 domains as fluorescence imaging tools, we expressed the clones as

fusions to eGFP in association with the expression regulating system developed for the abovementioned PSD95.FingR (*Gross et al., 2013*). The probe regulation is achieved by fusion of a transcription repressor and a zinc finger in combination with the incorporation of the corresponding zinc finger-binding motif upstream of the reporter gene in the expression plasmid (*Figure 1—figure supplement 3*). In this system, while eGFP is used to monitor the binding module and its target, the regulation system allows to avoid overexpression of the recombinant binder compared to its endogenous target.

We first investigated whether the AMPAR-mediated synaptic currents were affected by the presence of the various monobodies. Comparison of control eGFP-infected mouse neurons to neurons infected with Xph15, Xph18, or Xph20 did not reveal any significant difference on spontaneous mEPSCs (*Figure 2a–e*). The mean amplitude values were not modified by the presence of any of the PSD-95 binders (*Figure 2b*, control: 21.3 ± 2.1 pA [n = 19]; Xph15: 24.9 ± 3.9 pA [n = 15]; Xph18: 21.3 ± 2.0 pA [n = 17]; Xph20: 18.0 ± 2.2 pA [n = 16]; mean ± SEM with p>0.4 for the three clones using nonparametric Kruskal–Wallis test followed by Dunn's post hoc test). Similarly to the amplitude, nor the frequency (*Figure 2c*), the decay time (*Figure 2d*), or the rise time (*Figure 2e*) were affected as a result of the expression of the monobodies. We note that when similar measurements were performed in transfected rat hippocampal neurons none of the mEPSC parameters that we measured were significantly modified (*Figure 2—figure supplement 1*). We therefore conclude that expression of the three monobodies does not affect the AMPAR-mediated synaptic currents.

In addition to the electrophysiological measurements as an indicator of the proper synaptic recruitment of AMPARs, we also tested possible interference of the clones on the lateral mobility of surface AMPARs, as PSD-95 is the main AMPAR stabilizer (*Bats et al., 2007*). Transfected and non-transfected rat culture neurons were sparsely labeled in live condition with an ATTO-647N-conjugated antibody against the GluA2 subunit ectodomain. Single-particle tracking was performed by using the uPAINT method (*Giannone et al., 2010*) in order to gain insight on the AMPAR dynamics (*Figure 2f*). In agreement with the absence of modification of excitatory currents, no detectable effect was observed for Xph-expressing neurons vs control non-transfected ones on the lateral mobility of surface AMPARs. The distributions of diffusion coefficients were highly similar for all conditions (*Figure 2g*, *Figure 2—figure supplement 2*). Importantly, the percentage of mobile AMPARs was not increased in the presence of any of the clones as could have been expected from a binder that would have perturbed interactions with either of the first two PDZ domains (*Figure 2h*, control: 34.9 ± 9.5% [n = 26]; Xph15: 36.9 ± 11.9% [n = 27]; Xph18: 34.6 ± 9.9% [n = 18]; Xph20: 32.2 ± 10.5% [n = 7]; mean ± SD with p>0.73 by ordinary one-way ANOVA).

In complement to the assessment of a potential impact of the various Xph monobodies on one of the PSD-95 main partners, we also took a holistic proteomic approach to evaluate their effect on the entire PSD-95 interactome. In this case, we focused our investigation on Xph20, which together with Xph15 recognize the same PSD-95 epitope – but with a stronger affinity for Xph20 – and represent the most promising candidates for binding to PSD-95 with minimal impact on its function. Rat pyramidal neurons were either infected with eGFP or Xph20-eGFP (*Figure 2—figure supplement 3*) and lysed after 14 d of expression. PSD-95 was immunoprecipitated under mild conditions to preserve protein complexes and each sample was trypsin-digested and analyzed by LC-MS/MS. When comparing the two conditions, eGFP or Xph20-eGFP expression (*Figure 2i*), the abundance of most identified proteins was globally unaffected (abundance ratio close to 1 and/or low statistical significance with p-values above 0.05). All known PSD-95 partners that were identified in the two independent experiments (~40 identified partners per experiment) fell in that category. Less than 20 proteins showed both a clear change in abundance (absolute log2(abundance ratio) above 1.5) coupled to a significant p-value (below 0.05). None were reported as PSD-95 partner and most were only detected in only one of the two experiments. Only vimentin (Vim) and the glial fibrillary acidic protein (Gfap) were identified in both experiments; however, given their nature and function, the link to PSD-95 seems unlikely. In conclusion, in agreement with absence of detectable effect on AMPA receptors, the PSD-95 interactome appears unaffected by Xph20-eGFP expression.

Altogether, these experiments indicate that the binding of neither Xph15, Xph18, nor Xph20 affects endogenous PSD-95 function in its native environment as judged by the absence of impact on AMPAR properties and conservation of its interactome. These results are therefore consistent with the nature of each clone's respective epitope, which are found on regions of PSD-95 not involved in the PDZ domain binding of native cellular protein partners.

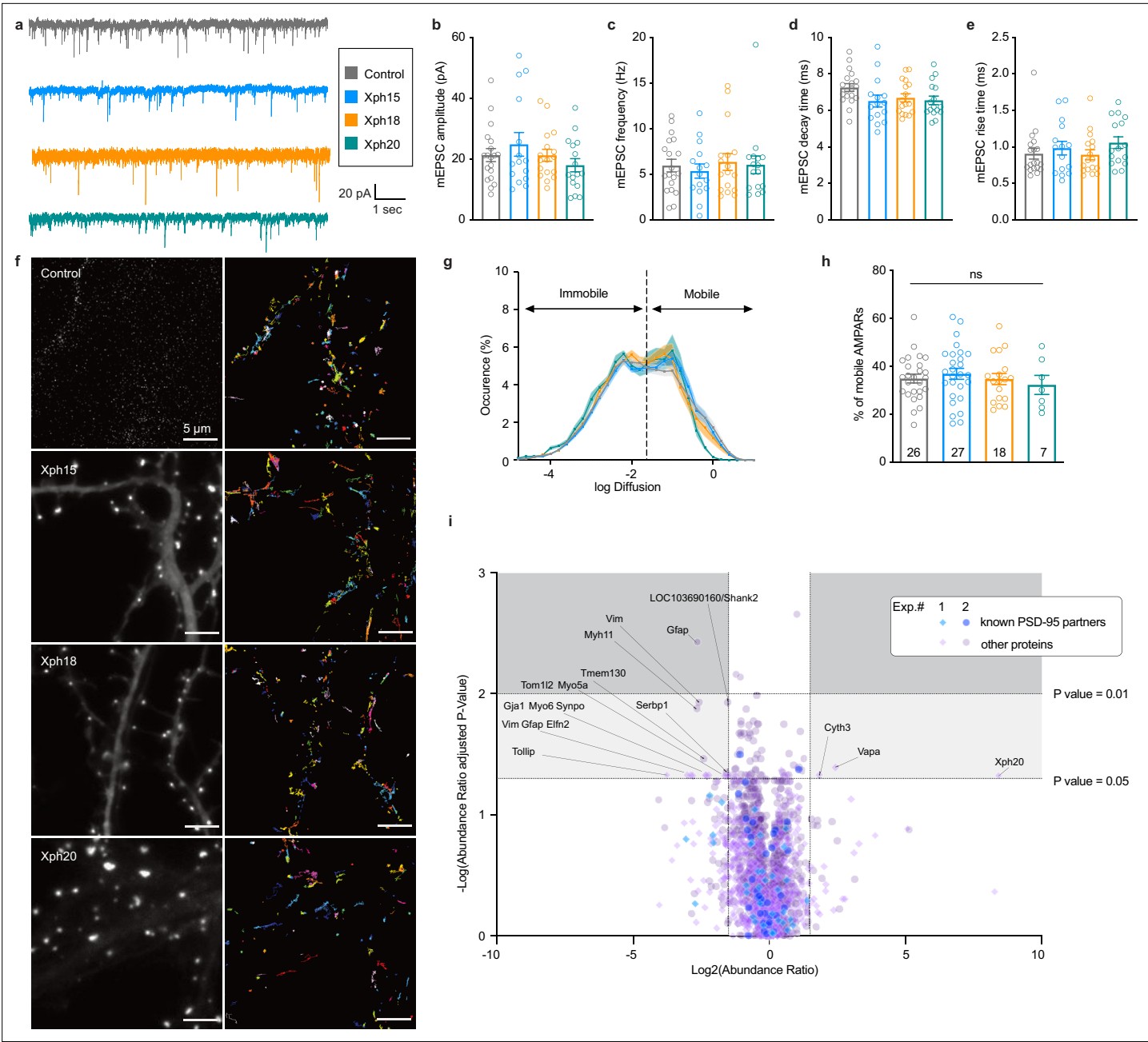

**Figure 2.** Evaluation of monobodies binding to endogenous PSD-95. (**a–e**) Synaptic currents in wild-type mouse neurons infected with adeno-associated viruses expressing either eGFP, Xph15, Xph18, or Xph20 (n = 19, 15, 17, and 16, respectively, from three independent cultures). (**a**) Representative traces of glutamatergic miniature excitatory postsynaptic currents (mEPSCs) recorded from neurons expressing eGFP, Xph15, Xph18, or Xph20, (**b**) mEPSC amplitude, (**c**) mEPSC frequency, (**d**) mEPSC decay time, and (**e**) mEPSC rise time. Data are expressed as mean ± SEM, statistic significances were tested using the nonparametric Kruskal–Wallis test followed by Dunn's post hoc test. Each dot represents a recorded cell. (**f**) Representative images for AMPARs single-particle tracking by uPAINT (left, epifluorescence image of non-transfected control and Xph15-, Xph18-, or Xph20-eGFP expression pattern in transfected rat neuron culture; right, trajectories; scale bar 5 μm). (**g**) Average distribution of instantaneous diffusion coefficients obtained by uPAINT of synaptic AMPAR with typical bimodal distribution. Error bars indicate cell-to-cell variability. (**h**) Percentage of mobile AMPARs (mean ± SEM, each dot represents the mean value of mobile AMPAR per cell). Statistical analysis was performed with an ordinary one-way ANOVA. (**i**) Evaluation of Xph20 expression on PSD-95 interactome. Volcano plot of the proteins identified by mass spectrometry following immunoprecipitation of endogenous PSD-95 in rat hippocampal culture infected by either Xph20-eGFP or eGFP (overlay of two independent experiments, experiment 1, diamonds; experiment 2, circles). Known PSD-95 partners are represented in blue shades while other proteins are represented in purple shades. Protein identity (gene name) is provided for those below a p-value of 0.05 and above an absolute log2(abundance ratio) value of 1.5.

*Figure 2 continued on next page*

*Figure 2 continued*

The online version of this article includes the following source data and figure supplement(s) for figure 2:

**Source data 1.** Spreadsheet with the miniature excitatory postsynaptic currents (mEPSCs) amplitude data (*Figure 2b*).

**Source data 2.** Spreadsheet with the miniature excitatory postsynaptic currents (mEPSCs) frequency data (*Figure 2c*).

**Source data 3.** Spreadsheet with the miniature excitatory postsynaptic currents (mEPSCs) decay data (*Figure 2d*).

**Source data 4.** Spreadsheet with the miniature excitatory postsynaptic currents (mEPSCs) rise time data (*Figure 2e*).

**Source data 5.** Spreadsheet with the diffusion distribution data (*Figure 2g*).

**Source data 6.** Spreadsheet with the percentage of mobile AMPARs data (*Figure 2h*).

**Source data 7.** The mass spectrometry proteomics data have been deposited to the ProteomeXchange Consortium via the PRIDE partner repository with the dataset identifier PXD045002 (*Figure 2i*).

**Figure supplement 1.** Spontaneous miniature excitatory postsynaptic currents properties based on the analysis of the 100 first events of control non-transfected or Xph15 and Xph18 (fused to eGFP and the expression regulation system) transfected rat culture neurons.

**Figure supplement 1—source data 1.** Spreadsheet with the miniature excitatory postsynaptic currents (mEPSCs) amplitude data (*Figure 2—figure supplement 1a*).

**Figure supplement 1—source data 2.** Spreadsheet with the miniature excitatory postsynaptic currents (mEPSCs) frequency data (*Figure 2—figure supplement 1b*).

**Figure supplement 1—source data 3.** Spreadsheet with the miniature excitatory postsynaptic currents (mEPSCs) decay time data (*Figure 2—figure supplement 1c*).

**Figure supplement 1—source data 4.** Spreadsheet with the miniature excitatory postsynaptic currents (mEPSCs) rise time data (*Figure 2—figure supplement 1d*).

**Figure supplement 2.** Cumulative distribution of average distribution of instantaneous diffusion coefficients obtained by uPAINT of synaptic AMPAR (*Figure 2g*).

**Figure supplement 2—source data 1.** Spreadsheet with the cumulative diffusion distribution data.

**Figure supplement 3.** Rat hippocampal culture transduced with adeno-associated viruses expressing either eGFP or Xph20-eGFP at 3 d in vitro (DIV) and imaged and lysed at 16 DIV for proteomics analysis (scale bar 40 μm).

## Evaluation of Xph15/18/20 as endogenous PSD-95 imaging probes

The absence of any detectable effect of Xph clone binding on PSD-95 function constituted an obligatory first criterion to consider their use as a non-interfering imaging probe. As the three monobodies comply with this criterion (albeit with some reservation for Xph18), we next focused on confirming their capacity to label endogenous PSD-95 and on evaluating their specific properties as fluorescent probes. First, we assessed the ability of Xph15, 18, and 20 to bind and target a fluorescent protein to PSD-95 in primary hippocampal neuron culture. Neurons were transfected with the previously tested Xph-eGFP fusions (or PSD95.FingR-eGFP [*Gross et al., 2013*], from which the expression vector was derived, as a comparison) chemically fixed after 23–27 days in vitro (DIV) and immunostained for PSD-95 (*Figure 3a*). For all the binders tested, the eGFP signal was similarly strongly enriched on dendrites at postsynapse-like structures. The objects we observed presented in all cases a mean intensity enrichment ratio compared to the rest of the dendrite around 7 (*Figure 3b*; PSD95.FingR: 6.4 ± 3.0; Xph15: 6.4 ± 1.4; Xph18: 7.4 ± 2.1; Xph20: 8.4 ± 2.1; mean ± SD with p>0.72 by ordinary one-way ANOVA followed by Tukey's multiple-comparison tests). This indicates that the four binders behave similarly in their capacity to address a fluorescent protein reporter to specific regions in neuronal cells. We next analyzed in each case how these objects colocalized with the labeling obtained by immunostaining of endogenous PSD-95 (*Figure 3a, c and d*). In general, colocalization percentage values ranged from 0 to 100, which we attribute to the inherent differences of the two staining methods being compared (i.e., expressed reporter vs antibody labeling post-fixation and permeabilization). The colocalization of PSD-95-positive objects detected by antibody immunostaining with the puncta revealed by the four investigated probes was overall strong (*Figure 3d*, PSD95.FingR: 75.4 ± 30.9; Xph15: 98.4 ± 4.8; Xph18: 95.5 ± 11.1; Xph20: 95.2 ± 13.4; mean ± SD with p<0.0001 for PSD95. FingR vs the other binders by ordinary one-way ANOVA followed by Dunnett's multiple-comparison test), in agreement with reported values for PSD-95.FingR (*Cook et al., 2019*; *Gross et al., 2013*). Remarkably, the values obtained for the three Xph clones were significantly higher than for PSD95. FingR. This trend was also visible when considering the colocalization between eGFP and antibody-stained objects, with Xph clones clearly showing a stronger enrichment in high colocalization values

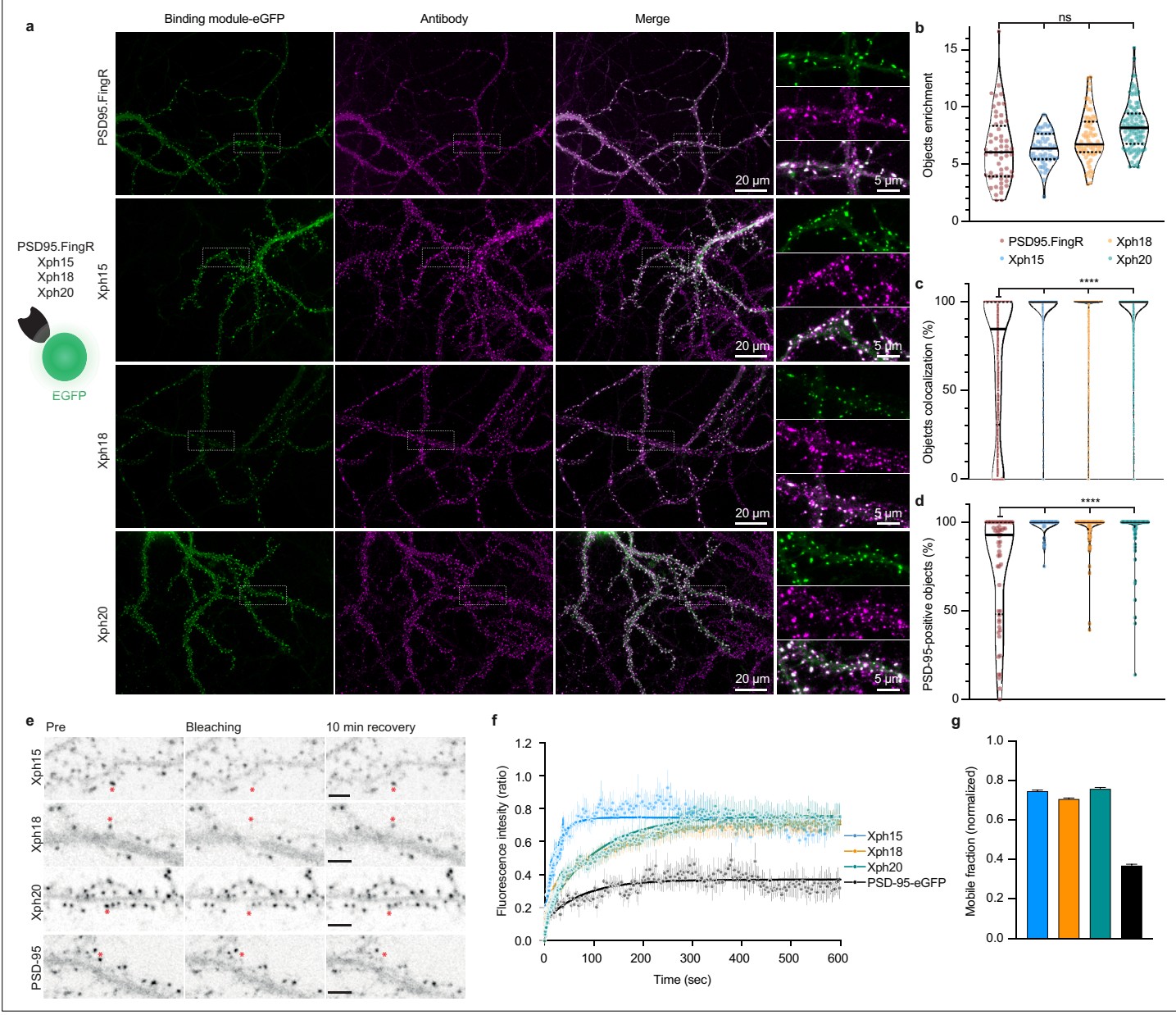

**Figure 3.** Evaluation of monobodies as intrabody fluorescent reporter probes. (**a**) Representative epifluorescence images of the eGFP-fused binding modules vs immunostaining of endogenous PSD-95 domain. For the zoomed regions, top: binding module; middle: antibody staining; bottom: merge. (**b**) Enrichment of object vs shaft fluorescence signal. Violin plots show median, first and third quartile, and all individual data points (each corresponding to the analysis of a single acquired image) pooled from at least three independent experiments. (**c**) Percentage of eGFP vs antibody objects colocalization (obtained by determining the percentage of common pixels within a probe labeled object with PSD-95 immunostaining). Violin plots show median, first and third quartile, and all individual data points (each corresponding to a detected object) pooled from at least three independent experiments. (**d**) Percentage of PSD-95-positive objects defined as objects with more than 50% pixel in common. Violin plots show median, first and third quartile, and all individual data points (each corresponding to the analysis of a single image) pooled from at least three independent experiments. (**e**) Representative images for fluorescence recovery after photobleaching (FRAP) experiments with eGFP fusion proteins, the red asterisk indicating the bleached dendritic spine. Scale bars 5 µm. (**f**) Fluorescence recovery analysis (mean ± SEM with fitted curve, n = 8/73, 10/108, 9/107, 5/77 cells/spines for Xph15, Xph18, Xph20, and PSD-95-eGFP, respectively, from at least two independent experiments). (**g**) Mobile probe fraction (mean ± SEM, n and color code same as **f**).

The online version of this article includes the following source data and figure supplement(s) for figure 3:

**Source data 1.** Spreadsheet with the raw object enrichment data (*Figure 3b*).

**Source data 2.** Spreadsheet with the raw object colocalization percentage data (*Figure 3c*).

*Figure 3 continued on next page*

*Figure 3 continued*

**Source data 3.** Spreadsheet with the raw PSD-95-positive objects percentage data (*Figure 3d*).

**Source data 4.** Spreadsheet with the fluorescence recovery data (*Figure 3f*).

**Source data 5.** Spreadsheet with the mobile fraction extracted from the fluorescence recovery data (*Figure 3g*).

**Figure supplement 1.** Regulated vs non-regulated expression of Xph20-FP fusion.

compared to PSD95.FingR (*Figure 3c*, PSD95.FingR: 65.8 ± 38.9; Xph15: 95.7 ± 15.8; Xph18: 92.3 ± 19.2; Xph20: 93.5 ± 16.8; mean ± SD with p<0.0001 for PSD95.FingR vs the other binders by ordinary one-way ANOVA followed by Dunnett's multiple-comparison test). We interpret this difference as a direct benefit from the specificity of the Xph clones for PSD-95 while PSD95.FingR, which can also bind SAP97 and SAP102, may also report to some extent the presence of these two paralog proteins. Considering the generally strong overlap of the GFP signal with the immunostaining of endogenous PSD-95, we conclude that the three monobodies label PSD-95 efficiently.

In order to evaluate the flexibility/versatility of the labeling system, we considered other fluorescent proteins, and in particular, a red fluorescent protein. We chose the recently described mScarlet-I as one of the brightest red reporters (*Bindels et al., 2017*). Despite several attempts, we failed at expressing the Xph20-mScarlet-I fusion in transfected cultured neurons as a result of a toxicity not observed for the eGFP constructs. Transfer of the Xph20-mScarlet-I fusion into a non-regulated plasmid resulted in non-toxic expression of the probe, albeit at a higher level compared to PSD-95 endogenous expression levels. It therefore led to a homogeneous filling of the whole neuron volume (*Figure 3—figure supplement 1*). This indicates that the toxicity is here a consequence of the association of mScarlet-I with the regulation system. Replacement of mScarlet-I with another bright monomeric red fluorescent protein, mRuby2 (*Lam et al., 2012*), abolished the observed toxic effect and provided a similar staining compared to Xph-eGFP fusions (*Figure 3—figure supplement 1*).

While these surprising results suggest that not all fluorescent proteins are compatible with the expression regulation system, they also highlight the critical need to match the target expression levels for imaging applications. In particular, as reported for PSD95.FingR, the expression regulation system applied to the Xph binders allows for long expression schemes without excessive or detectable over-production of the probe. This possibility in turn provides flexibility to handle the timing of the genetically encoded probe delivery without compromising the achieved labeling steady state.

The binding kinetics of the Xph clones previously evaluated by surface plasmon resonance (SPR) showed different but overall rather fast association and dissociation rate constants indicating fast exchanging complexes (half-lives of ~2, 28, and 10 s for Xph15, Xph18, and Xph20, respectively, for the isolated recombinant PSD-95 PDZ domains 1 and 2) (*Rimbault et al., 2019*). These kinetic profiles were also associated with moderate affinities with binding constants in the low micromolar (4.3 and 2.6 µM for Xph15 and 18, respectively) to sub-micromolar range (330 nM for Xph20). We therefore sought to further evaluate how these properties would translate in the context of their use as PSD-95 labeling tools. To this end, we used fluorescence recovery after photobleaching (FRAP) to determine how the different probes interact with their target in its native environment. Fluorescence recovery was measured in photobleached single synapses (Xph objects) in neurons expressing the Xph-eGFP fusions (*Figure 3e*). As expected, on the minute timescale, the three monobodies showed fast exchange rates (*Supplementary file 1*) as well as high mobile fractions (*Figure 3f* and *Supplementary file 1*; 75, 71, and 76% for Xph15, Xph18, and Xph20, respectively) compared to the values reported in basal conditions for PSD-95-GFP knock-in (~10% after 60 min [*Fortin et al., 2014*]) or to the values obtained here with a transfected PSD-95-eGFP (37%). The measured mobile fractions and the half-lives (~16, 71, and 70 s for Xph15, Xph18, and Xph20, respectively) are consistent with the SPR kinetics measurements, with Xph15 being the fastest and Xph18 the slowest. We note that the results we obtained here for the probes account for the behavior of both the free and the PSD-95-bound populations. However, considering the large difference between the values obtained for the probes and for PSD-95, we can reasonably conclude that the Xph-derived probes exchange and are being renewed at a faster rate than their target.

We have previously shown that the three selected monobodies were specific binders of PSD-95 with no detectable interaction with its paralogs (PSD-93, SAP97, and SAP102). Structural and binding investigations on isolated tandem PDZ domains as well as full-length proteins in model cell line were

consistent with the recognition of an epitope unique to PSD-95 (around key residue F119, which is replaced by an arginine in all other paralogs). In order to confirm that these advantageous properties were conserved when used as intrabodies to label endogenous PSD-95 in its native environment, we used a knock-down approach with a short hairpin RNA (shRNA)-targeted against PSD-95 (*Schlüter et al., 2006*). We focused here on the two most promising binders, Xph15 and Xph20. Hippocampal rat neurons were transfected with the anti-PSD-95 shRNA or a control scramble shRNA together with either Xph15-eGFP or Xph20-eGFP. Neurons were fixed after 4–5 d of expression and immunostained for PSD-95. Our experimental conditions were designed in order to avoid potential issues associated with shRNA overexpression and were characterized by heterogeneity in the levels of PSD-95 knock-down. While the conditions with the scramble shRNA led to a labeling very similar to what was obtained previously in its absence, the use of the shRNA against PSD-95 was associated with a clear and consistent loss of signal, in particular no detectable puncta, both for the two Xph intrabodies and the corresponding antibody labeling (*Figure 4a–c*). PSD-95 knock-down effect manifested itself by an eGFP fluorescence signal almost exclusively nuclear indicating the absence of cytosolic target for intrabodies (*Figure 4a*) and was quantitatively observed both in detected objects (*Figure 4b*) and over entire dendritic segments (*Figure 4c*) demonstrating the specific recognition of endogenous PSD-95 by Xph15 and Xph20.

The PSD-95 downregulation experiment led to a clear loss of Xph15 and Xph20 dendritic labeling, suggesting a direct correlation between PSD-95 expression levels and the two intrabodies' cytosolic distribution. To further characterize this correlation, we co-expressed Xph15-eGFP or Xph20-eGFP with a PSD-95 mScarlet-I fusion (*Figure 4d*). Systematic analysis of the two fluorescence signals (eGFP and mScarlet-I) indicated strong correlation, with Pearson correlation coefficients of 0.84 and 0.87 for Xph15 (*Figure 4e*) and Xph20 (*Figure 4f*), respectively. These results illustrate the combined efficiency of the two Xph and the regulation system to match the intrabodies' levels to those of PSD-95.

Overall, this ensemble of results demonstrates first that Xph15, Xph18, and Xph20 can be used to efficiently recognize and bind endogenous PSD-95 with minimal impact on its function. In addition, the fusion of a fluorescent protein to the monobodies (together with the use of an expression regulation system) allows in this context to dynamically label PSD-95 in live conditions. The resulting labeling of PSD-95 is clearly specific and correlates strongly with the levels of the target protein, as demonstrated for both Xph15 and Xph20. Considering the large epitope recognized by Xph18, which as we have shown leads to a constrained conformation of the tandem PDZ domains, we chose here to only focus on Xph15 and Xph20 that both recognize a smaller epitope restricted to PDZ 1 to further develop the imaging tools and fully ensure minimal interference of the resulting probes.

## Engineering probes for super-resolution imaging

The previous experiments validated the use of Xph-derived constructs as imaging probes to monitor endogenous PSD-95. The specific recognition properties of the evolved $^{10}$FN3 domains coupled to the capacity to match their expression levels to those of PSD-95 therefore provide an ideal platform to further elaborate our clones into more advanced probes, in particular for SRI applications. To this end, we modified the GFP reporter part of the probes with systems better adapted for various SRI modalities.

We first investigated how the probes performed with STED microscopy. STED is a point-scanning method that relies on the simultaneous use of both an excitation and a depletion laser beam (*Sahl et al., 2017*; *Vicidomini et al., 2018*). Since the technique is compatible with a number of fluorescent proteins, its implementation is here relatively straightforward. A first attempt to determine whether expression levels were compatible with STED imaging was performed on fixed cultured neurons expressing Xph20-eGFP. Without the need to improve the fluorescent protein part, the results were satisfactory with a clear gain in resolution when comparing STED and confocal imaging (*Figure 5— figure supplement 1a and b*).

In comparison to other imaging techniques, one of the main advantages of STED is the compatibility with live imaging, in particular for dynamic studies. While alternative methods exist to label endogenous PSD-95 post-fixation/permeabilization, live labeling of PSD-95 still remains a challenge for which Xph15 or Xph20 can provide solutions. A major drawback of STED is the high illumination intensities required in particular for efficient depletion that often results in photobleaching of the

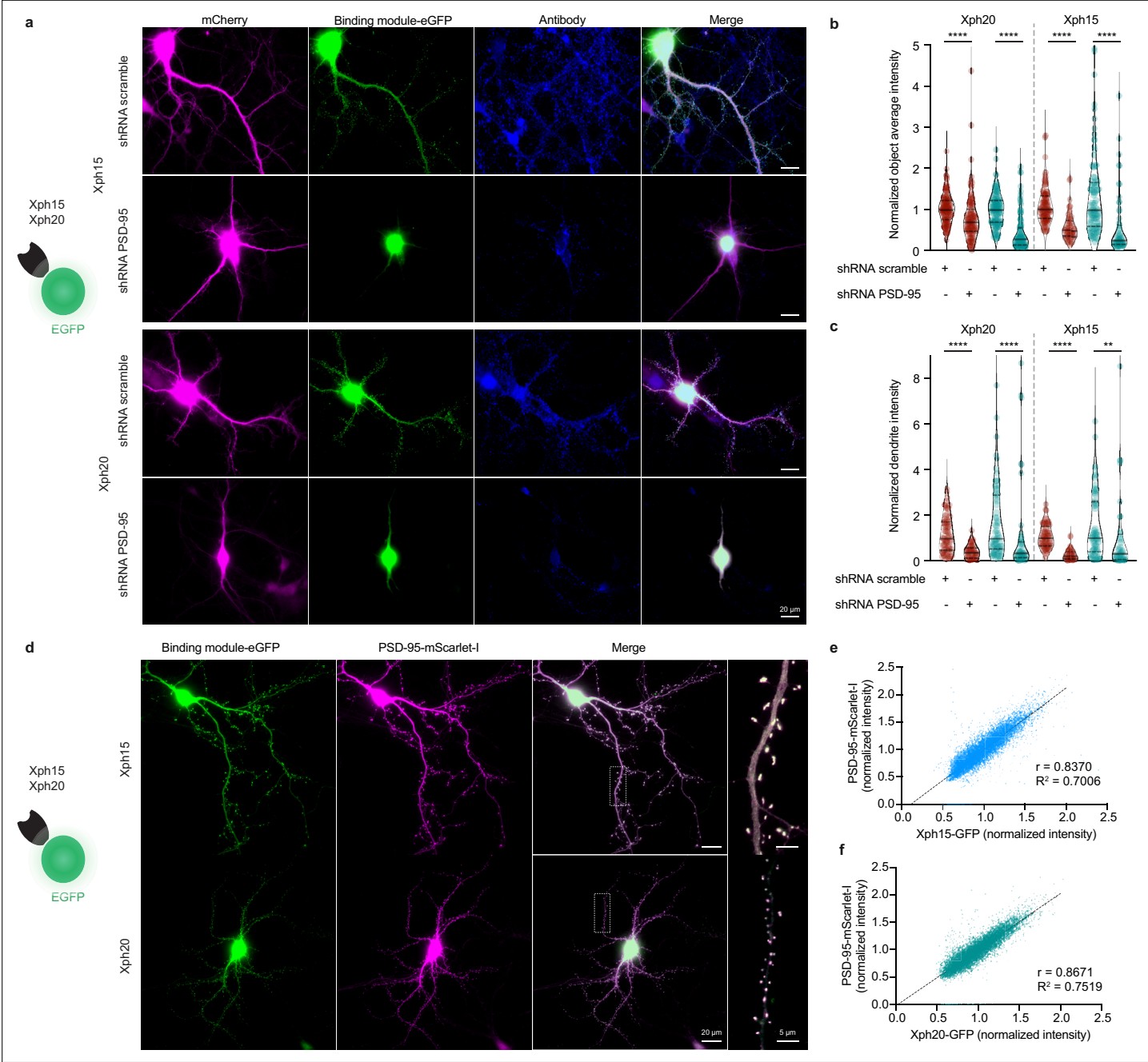

**Figure 4.** Evaluation of Xph15 and Xph20 intrabodies specificity. (**a**) Representative epifluorescence images showing Xph15-eGFP (top) and Xph20-eGFP (bottom) expressed in pyramidal neuron together with a scramble shRNA or the shRNA against PSD-95 both associated to a mCherry soluble fluorescent reporter (magenta). The neurons are fixed and immunostained for PSD-95 (blue). (**b**) Comparison of objects average fluorescence intensity (red: immunostaining fluorescence signal; teal: Xph-eGFP signal) on neurons transfected with the scramble shRNA or the shRNA against PSD-95. Normalization was performed by using the median of the scramble shRNA condition. Violin plots show median, first and third quartile, and all individual data points (each corresponding to the analysis of a single detected object) pooled from three independent experiments, statistic significances were tested using the nonparametric Kruskal–Wallis test followed by Dunn's post hoc test. (**c**) Comparison of dendrites fluorescence intensity (red: immunostaining fluorescence signal; teal: Xph-eGFP signal; integrated fluorescence signal per area units) on neurons transfected with the scramble shRNA or the shRNA against PSD-95. Normalization was performed by using the median of the scramble shRNA condition. Violin plots show median, first and third quartile, and all individual data points (each corresponding to the analysis of a single dendritic fragment) pooled from two independent experiments, statistic significances were tested using the nonparametric Kruskal–Wallis test followed by Dunn's post hoc test. (**d–f**) Correlation of Xph15 and Xp20 intrabodies with PSD-95. (**d**) Representative immunofluorescent images showing Xph15-eGFP (top), Xph20-eGFP (bottom), and PSD-95-mScarlet-I. The dendritic region within the white box is enlarged below to better illustrate colocalization the eGFP and mScarlet-I signal. Scatter plots

*Figure 4 continued on next page*

Figure 4 continued

showing the correlation between Xph15- (**e**) or Xph20-eGFP (**f**) and PSD-95-mScarlet-I normalized fluorescence intensity. Pearson correlation coefficients and slopes (simple linear regression) between eGFP and mScarlet-I fluorescent intensity were calculated using GraphPad Prism 8. Data from 67 dendrites and 8362 synapses for Xph15, and 63 dendrites and 8004 synapses for Xph20 (two independent experiments).

The online version of this article includes the following source data for figure 4:

**Source data 1.** Spreadsheet with the normalized object average intensity data (*Figure 4b*).

**Source data 2.** Spreadsheet with the normalized dendrite intensity data (*Figure 4c*).

**Source data 3.** Spreadsheet with the normalized fluorescence intensity data (*Figure 4e*).

**Source data 4.** Spreadsheet with the normalized fluorescence intensity data (*Figure 4f*).

fluorophore. In this context, the fast-exchanging properties of the probes could be an advantage and allow, by fast renewal of the probes, repeated acquisitions of the same region.

For live experiments, we therefore used the fastest exchanging binder, Xph15, to fully benefit from maximal probe replacement. In parallel, the fluorescent protein eGFP was replaced by mNeonGreen (*Shaner et al., 2013*) for its higher quantum yield and improved photostability. The Xph15-derived probe expressed well and provided a labeling similar to the eGFP construct in live dissociated neurons as judged by confocal microscopy (*Figure 5a–c*). Despite the improved properties of mNeonGreen, application of a STED illumination invariably led to significant photobleaching of the area investigated (*Figure 5b*). Nevertheless, the imaged area was repopulated over time with fresh probes as could be anticipated from the FRAP experiments. About 70% of the initial fluorescence was recovered in less than 15 min (*Figure 5d*), allowing the area to be efficiently imaged repetitively. We note however that while the confocal imaging quality was comparable to the one obtained prior to the STED imaging, in that timescale, the STED quality was still noticeably degraded due to the loss of signal. Avoiding the repeated confocal imaging as well as reducing the area of STED imaging could be simple strategies to further improve the fluorescence recovery by limiting the photobleaching associated with unnecessary light exposure and by locally increasing the pool of intact probes vs photodamaged ones.

In order to more efficiently circumvent the loss of signal and enable faster repeated STED acquisitions, for instance, with 3D-stacks or minute-timescale super-resolution investigations, we modified our strategy and opted for the use of brighter and more photoresistant organic dyes. To effectively functionalize our probes with such dyes, we replaced the fluorescent protein by a SNAP-tag (*Keppler et al., 2003*; *Figure 5e*) and used a cell-permeant silicon rhodamine fluorophore (SiR) (*Lukinavičius et al., 2013*) coupled to benzylguanine added prior to the imaging session. The SNAP-tag probe behaved comparably to the eGFP version, and synaptic objects, hallmark of PSD-95 neuronal distribution, could be visualized (*Figure 5f*). Efficiency of the STED imaging was improved by the use of the brighter SiR dye (*Figure 5f and g*, *Figure 5—figure supplement 1c and d*). Photostability and dynamic exchange of the probe allowed to perform timelapse acquisitions at about a 1 min (50 s) frequency with minimal impact on the STED signal (*Figure 5g*), thereby illustrating the advantage of organic dyes over fluorescent proteins for such applications.

Single-molecule localization microscopy (SMLM) is another strategy used to access spatial resolution below the limit imposed by the diffraction of light (*Sauer and Heilemann, 2017*). It relies on temporal decorrelation of fluorophore emissions to obtain sparsely located fluorescent entities while keeping the majority of the population in non-emissive states. One strategy to perform SMLM is photoactivation localization microscopy (PALM) based on the use of photoactivatable or photoconvertible fluorescent proteins. To implement this imaging modality, we thus replaced eGFP with the monomeric photoconvertible protein mEos3.2 (*Zhang et al., 2012*). We considered Xph20 as the binding module for its stronger affinity and slower off-rate kinetics. The photoconvertible probe was expressed in dissociated cultured neurons and provided in its basal green state a labeling similar to the eGFP probe (*Figure 6a and b*).

Stochastic photoconversion of mEos3.2 was first performed in fixed neurons, and analysis of the resulting image stacks used to generate super-resolved images (*Figure 6b*). Efficiency of the fixation step on the probe was assessed by determination of the diffusion coefficients distribution of the detected single emitters. The results confirmed that a large majority of the investigated emitters were immobile (*Figure 6—figure supplement 1a–d*). The reconstructed maps showed a clear enrichment of the probe at synapses as already observed with diffraction-limited imaging techniques and STED.

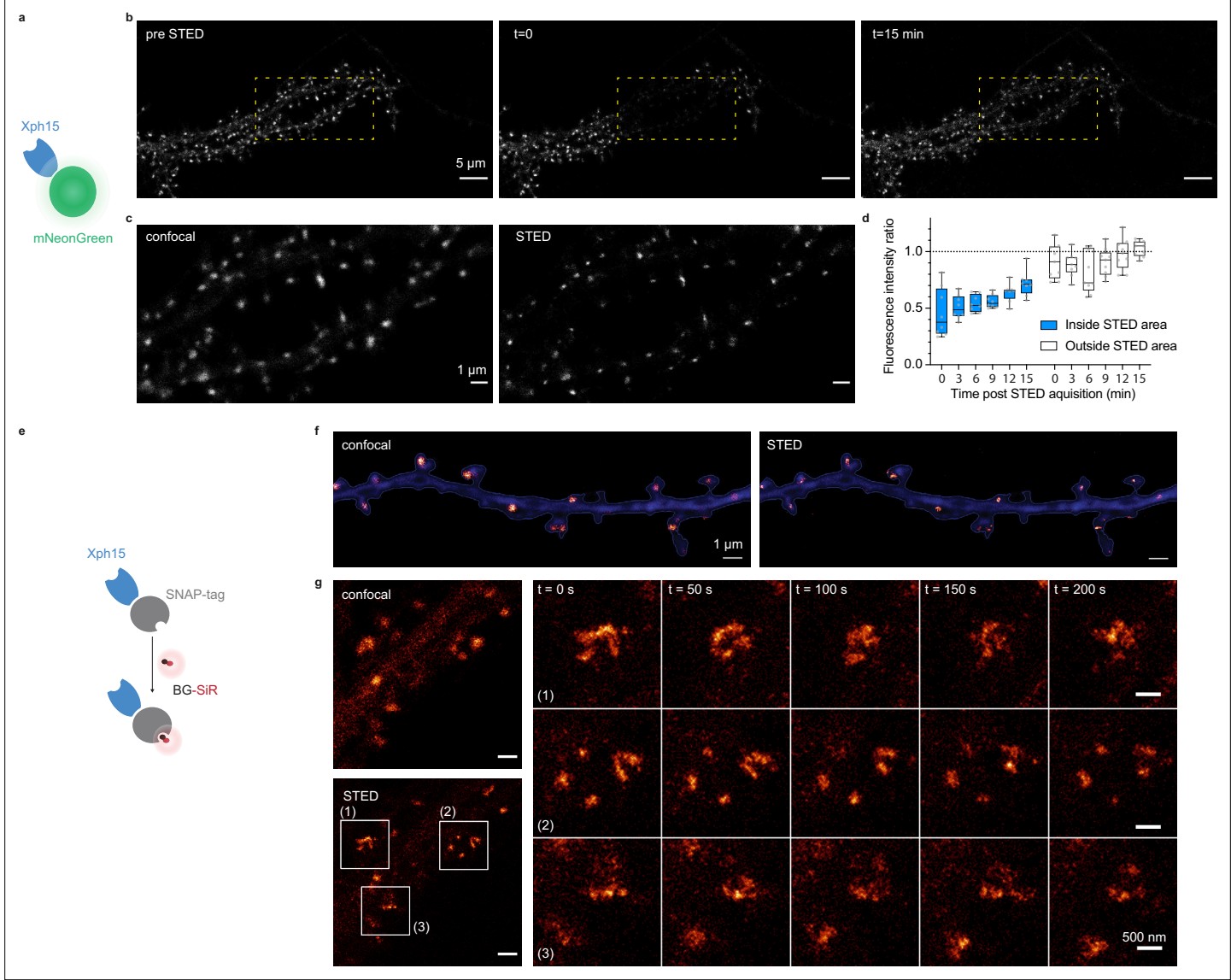

**Figure 5.** Evaluation of probes for stimulated emission depletion (STED) imaging. (**a**) Schematic representation of fluorescent protein-fused STED probe. (**b**) Representative confocal images of a neuron transfected with Xph15-mNeonGreen before and after STED. The yellow box corresponds to the STED region. (**c**) Confocal and STED images of the yellow box region from (**b**). (**d**) Evolution of fluorescence intensity over time of fluorescent objects subjected or not to STED (n = 8 and 9 for regions outside and inside of STED area, respectively). Box plots show median, first and third quartile, with whiskers extending to the minimum and maximum and all individual data points (each corresponding to a single object) pooled from at least two independent experiments. (**e**) Schematic representation of SNAP-tag-fused STED probe with the BG-SiR fluorophore. (**f**) Confocal and STED images of a neuron transfected with Xph15-SNAP-tag after incubation with BG-SiR. (**g**) Time course of repeated STED acquisitions with Xph15-SNAP-tag/BG-SiR.

The online version of this article includes the following source data and figure supplement(s) for figure 5:

**Source data 1.** Spreadsheet with the fluorescence intensity ratio data (*Figure 5d*).

**Figure supplement 1.** Stimulated emission depletion (STED) imaging.

Further analysis of the synaptic objects using the Tesselation-based segmentation method (*Levet et al., 2015*) revealed a non-homogeneous distribution with the presence of higher density clusters. The clusters represented about half the number of detections measured for the whole synaptic objects. A tentative estimation of single-emitter contribution (*Figure 6d*, *Figure 6—figure supplement 1e*, median ~10 detections) suggests the presence of ~200–300 probe copies per synaptic objects. This value is consistent with reported estimations of PSD-95 synaptic copies (*Chen et al., 2005*; *Sugiyama et al., 2005*) and therefore suggests a close to stoichiometric labeling of the

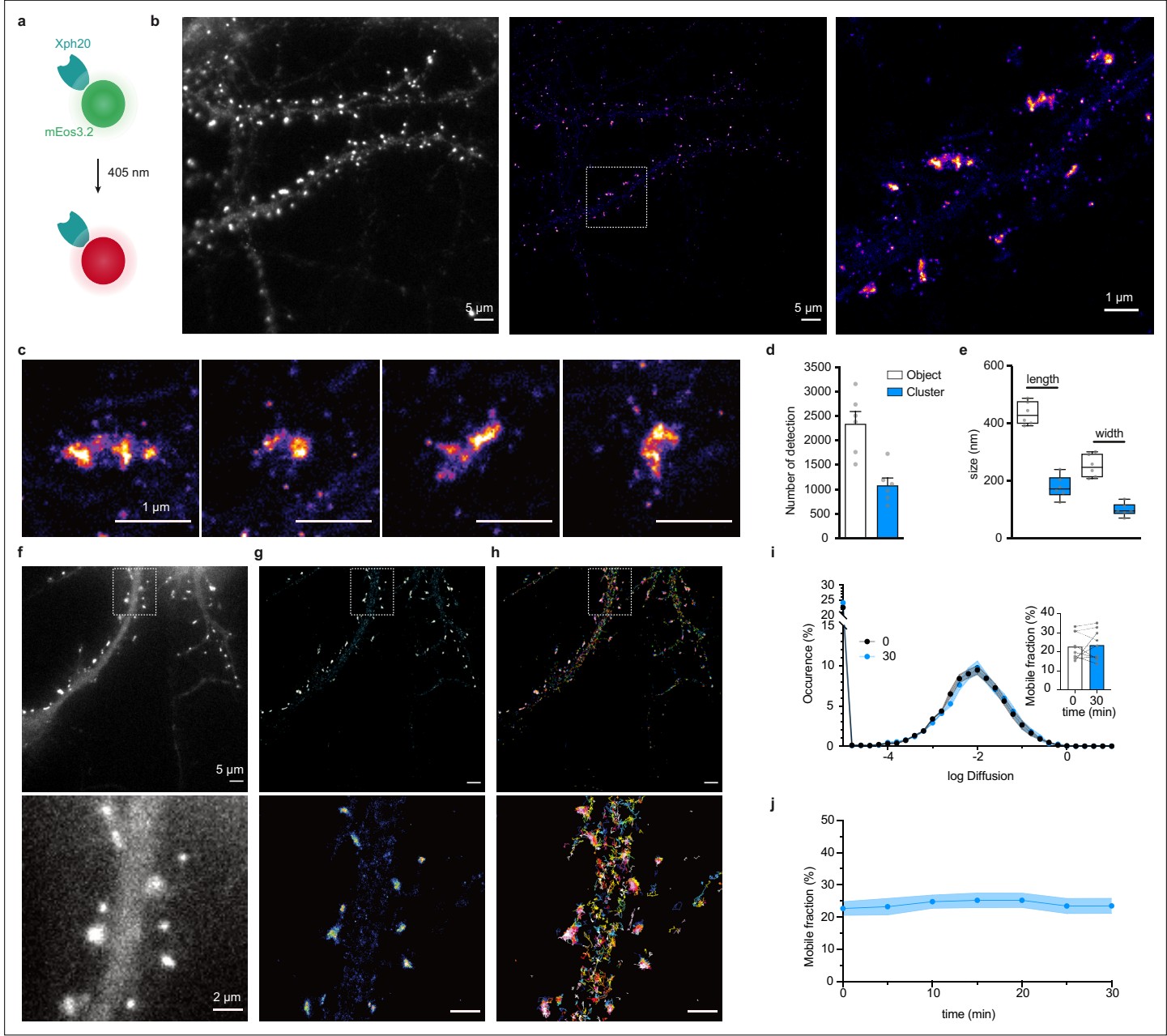

**Figure 6.** Evaluation of mEos3.2-derived probes for photoactivation localization microscopy (PALM) and spt-PALM applications. (**a**) Schematic representation of mEos3.2-fused probe. (**b**) Representative epifluorescence and PALM images of Xph20-mEos3.2 in fixed culture neurons. Left: epifluorescence image obtained from the native non-photoconverted green form of mEos3.2; middle: super-resolution image obtained by PALM from a sequence of 20,000 images of sparse single molecules of the photoconverted red from of mEos3.2 (scale bar 5 µm); right: zoomed region (scale bar 1 µm). (**c**) Examples of individual synapses showing PSD-95 organization at the postsynaptic density ('object') and sub-synaptic domain ('cluster'). Scale bar 1 µm. (**d**) Number of detections in 'objects' vs 'clusters' (mean ± SEM, each data point represents a single neuron). (**e**) Morphological analysis of 'objects' and 'clusters' (mean ± SEM, each data point represents a single neuron). (**f–h**) Representative epifluorescence and spt-PALM images of live culture neurons expressing Xph20-mEos3.2. epifluorescence of the expressed probe (before photoconversion) (**f**), super-resolution intensity map obtained by sptPALM from a sequence of 4000 images of sparse single molecules of the photoconverted red from of mEos3.2 (**g**), and trajectories of PSD-95 tagged with Xph20-mEos3.2 (**h**). Scale bars 5 and 2 µm for top and bottom images, respectively. (**i**) Average distribution of instantaneous diffusion coefficients obtained by spt-PALM of PSD-95 labeled with Xph20-mEos3.2 (at 0 min, $t_0$, beginning of the imaging session) or after 30 min of imaging ($t_{30}$). Error bars indicate cell-to-cell variability. Inset: percentage of the mobile fraction of probes at $t_0$ vs $t_{30}$ (mean ± SEM, each dot represents a single cell, n = 10). (**j**) Time course of the percentage of mobile probes (mean ± SEM, each dot represents a single cell, n = 10).

The online version of this article includes the following source data and figure supplement(s) for figure 6:

**Source data 1.** Spreadsheet with the number of detection data (*Figure 6d*).

*Figure 6 continued on next page*

*Figure 6 continued*

**Source data 2.** Spreadsheet with the length and width data (*Figure 6e*).

**Source data 3.** Spreadsheet with the diffusion distribution data (*Figure 6i*).

**Source data 4.** Spreadsheet with the mobile fraction percentage over time data (*Figure 6j*).

**Figure supplement 1.** Evaluation of mEos3.2-derived probes for photoactivation localization microscopy (PALM) applications.

**Figure supplement 1—source data 1.** Spreadsheet with the diffusion distribution data (*Figure 6—figure supplement 1c*).

**Figure supplement 1—source data 2.** Spreadsheet with the mobile fraction percentage data (*Figure 6—figure supplement 1d*).

**Figure supplement 1—source data 3.** Spreadsheet with the frequency distribution of single detection data (*Figure 6—figure supplement 1e*).

endogenous protein. Morphological analysis of the objects and clusters provided dimensions consistent with previous reports for PSD-95-mEos2 fusions with PALM (*Nair et al., 2013*) or by STORM by labeling endogenous PSD-95 with antibodies (*Compans et al., 2019*) (length: 434.5 ± 39.9 and 178.0 ± 39.3 nm; width: 251.1 ± 39.2 and 99.1 ± 22.4 nm for the objects and clusters, respectively, *Figure 6e*). Together, these results indicate that Xph20-mEos3.2 provides an accurate snapshot of PSD-95 nanoscale distribution in fixed samples.

PALM can also be performed on live samples, and single-particle tracking approaches yield in this case information on protein dynamics. This approach is typically achieved with a direct genetic fusion between the protein of interest and a photoconvertible fluorescent protein. Considering the efficiency of the evolved ${}^{10}$FN3 domain-mediated labeling, we investigated here how this approach could be implemented with Xph20-mEos3.2. Indeed, single emitters are tracked on a timescale over an order of magnitude faster (~500 ms) than the probe-target exchange dynamics (half-life of ~10 s), which should avoid bias linked to the occurrence of particles alternating between PSD-95 bound and unbound states.

Tracked particles were detected within the whole dendrite (*Figure 6f–h*). As observed previously with other imaging techniques, a strong enrichment of the probe was observed at the synapse when reconstructing the intensity maps corresponding to the accumulation of track coordinates. The probe diffusion coefficient showed a Gaussian-like distribution, which suggests a homogeneous population, with ~80% of particles confined or immobile and only ~20% mobile (*Figure 6i*). These results are highly similar to those obtained with a mEos2-fused (*Chazeau et al., 2014*) or a mVenus-fused PSD-95 (*Fortin et al., 2014*), suggesting that, in these conditions, we are essentially detecting probes bound to PSD-95. Indeed, a freely diffusive emitter, such as an unbound probe, would be characterized by faster diffusion coefficients (*Chazeau et al., 2014*) that could not be detected in these experimental conditions. Importantly, single-particle-tracking-PALM measurements could be repeated over the course of 30 min without detectable differences in the diffusion distribution (*Figure 6j*), demonstrating that the method is robust and compatible with hour timescale live investigations such as needed for synaptic plasticity events.

Considering the compatibility of our probes with STED and PALM, we next investigated their implementation to more recent SRI techniques adapted to the detection of multiple distinct targets. DNA-PAINT (*Jungmann et al., 2014*) constitutes a powerful alternative approach to STORM or PALM for SMLM, in particular for multiplexing applications, as it allows sequential imaging of different proteins of interest with the same fluorophore. The technique is based on the use of pairs of short complementary oligonucleotides, one strand bound to a target or its probe (docking strand) and the other functionalized with a fluorophore (imager strand), that undergo fast dynamic exchange between the bound and unbound states. In order to couple the docking strand to the Xph-derived probes, we considered here the use of either SNAP- or HaloTag (*Los et al., 2008*) to enzymatically create a covalent bound with benzylguanine- or haloalkane-derived oligonucleotides (*Schlichthaerle et al., 2019*). For the binding module, as for PALM, we chose Xph20 for its stronger affinity.

Each construct was co-transfected with a soluble eGFP marker in dissociated culture hippocampal neurons and used to implement the DNA-PAINT method after chemical fixation. The self-labeling tags were each reacted with the corresponding docking strand and after removal of the excess material, the Cy3B-derived imaging strand was applied to the samples. A control experiment in which no docking strand was added confirmed the very low level of non-specific binding of the imaging strand in our conditions (*Figure 7—figure supplement 1a*). For the HaloTag fusion, a homogeneous staining of the transfected neurons was observed (*Figure 7—figure supplement 1b*), suggesting a failure

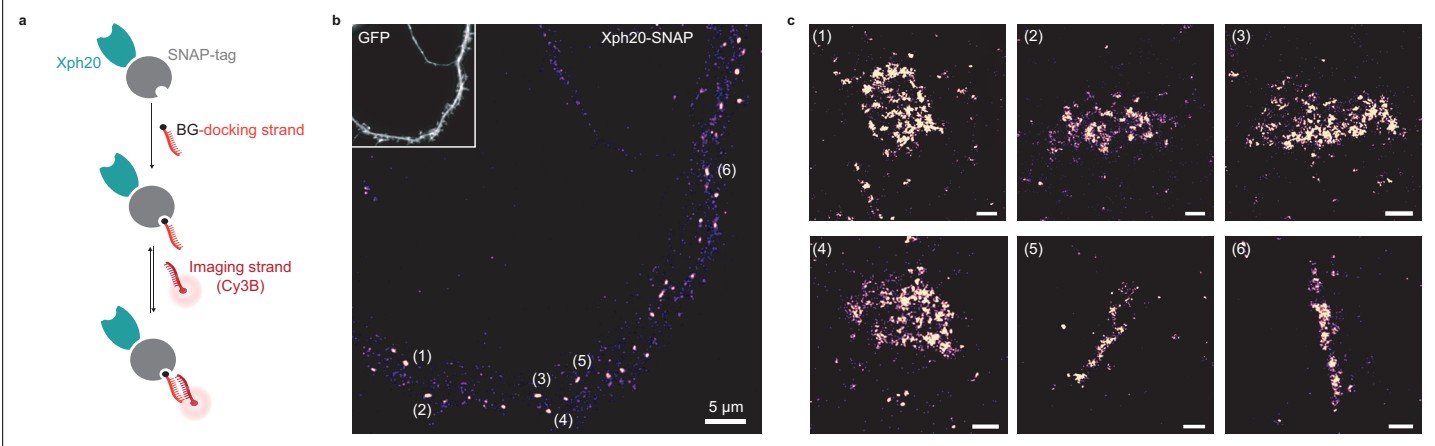

**Figure 7.** Evaluation of SNAP-tag-derived probes for DNA-PAINT super-resolution microscopy. (**a**) Probe design and labeling scheme (BG: benzylguanine). (**b**) Reconstructed DNA-PAINT image (10 Hz, 32,000 frames) of Xph20-SNAP-tag in the dendrites of a 14 day in vitro (DIV) hippocampal primary neuron (inset corresponding to soluble GFP). (**c**) Magnified views of the regions marked in (**b**) (scale bars 100 nm).

The online version of this article includes the following figure supplement(s) for figure 7:

**Figure supplement 1.** DNA-PAINT imaging.

either of the target recognition or of the regulation system with this particular engineered enzyme. The reason for this failure was not investigated further. We note that the larger size of the HaloTag (34 kDa vs 20 or 27 kDa for the SNAP-tag and fluorescent proteins, respectively) might impair efficient nuclear entry of the excess fusion protein product. In contrast, and consistently with the STED experiments, the SNAP-tag fusion allowed an efficient labeling with a clear synaptic enrichment comparable to the ones obtained with the other validated fusions (*Figure 7*).

Altogether, these results demonstrate that Xph15 and Xph20 constitute robust and valuable modules to engineer SRI probes for endogenous PSD-95. Indeed, by adapting the recognition and the reporting modules together with the use of a system for regulation of the probe production, we show that they can be easily exploited to provide a straightforward access to both the nanoscale mapping and the dynamics of this key synaptic protein.

## Targeting sensors to synapses

With the series of fluorescent or self-labeling protein fusions to Xph15 and 20, we have demonstrated the efficiency of the evolved binders to be used as the targeting module to report on the localization of endogenous PSD-95. Considering the highly enriched distribution of PSD-95 at excitatory postsynapses, we sought to expand the scope of application of Xph15/20 by exploiting their binding properties to target sensors or bioactive proteins at the postsynapse.

To validate this strategy, we used the genetically encoded calcium reporter GCaMP (*Chen et al., 2013*; *Dana et al., 2019*) with the aim to generate a direct fluorescent indicator of individual synapse activity (*Figure 8a*). A first attempt with GCaMP6f (*Chen et al., 2013*) as simple fusion to Xph15 expressed in rat primary culture neurons clearly indicated the feasibility of the approach (*Figure 8—figure supplement 1a and b*, *Figure 8—videos 1 and 2*). Indeed, even in the absence of the regulation system, a clear synaptic enrichment of the engineered calcium reporter was observed in comparison to the original sensor expressed alone. Expression levels were consistently low for the engineered construct, which can explain why the regulation system was not needed here to prevent excess probe production. We next attempted to improve the tool by using Xph20 as a stronger binder, GCaMP7f (*Dana et al., 2019*) as a brighter reporter, a stronger promoter (CAG instead of CMV), as well as the expression regulation system.

The two modified reporters (with and without the expression regulation system) were co-expressed with Homer-DsRed as a synaptic marker. Expression levels of the reporter were higher with the CAG promoter both in the absence and presence of the regulation system. However, in this case, the latter was necessary to allow a clear synaptic enrichment of GCaMP7f (*Figure 8b–d*) as its absence, combined with higher expression levels, led to a homogeneous distribution of the calcium reporter in

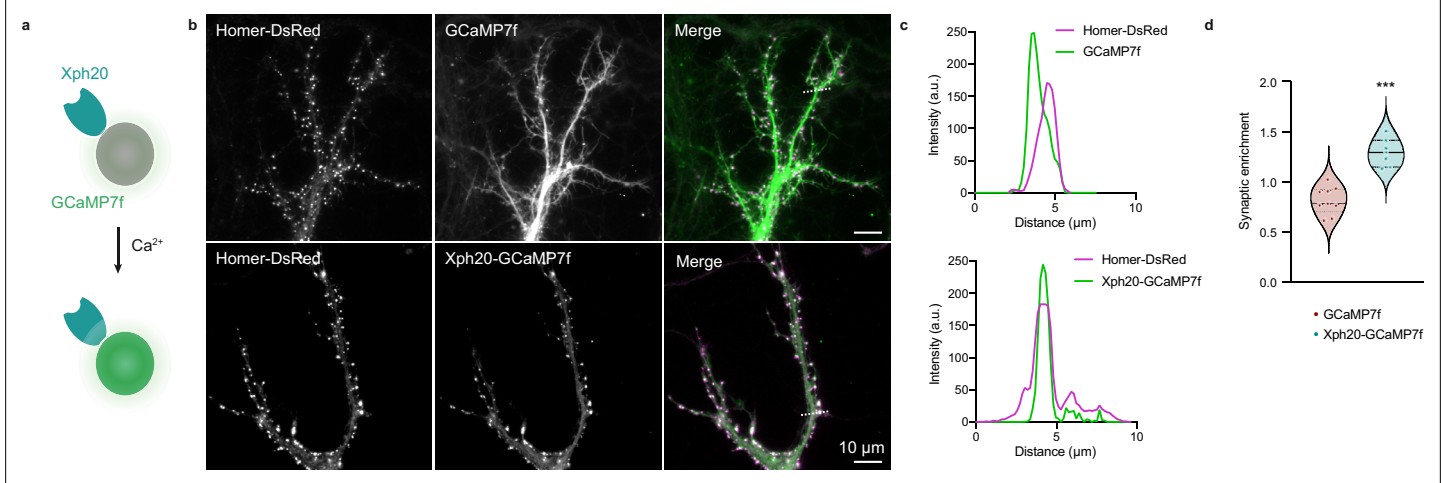

**Figure 8.** Application of the ReMoRa method for the synaptic targeting of calcium reporters. (**a**) Schematic representation of calcium signaling probe. (**b**) Comparison of the expression profile of targeted and regulated (Xph20-GCaMP7f, bottom panel) vs parental calcium sensor (GCaMP7f, top panel) for GCaMP7f synaptic targeting (GCaMP in green and Homer-DsRed in magenta in the merged images). (**c**) Linescans from (**b**) comparing the probe repartition between shaft and spine compartments. The linescans show a clear enrichment of the regulated probe in neuronal spines. (**d**) Probes synaptic enrichment determined using Homer-DsRed staining as a synaptic marker (n = 9 and 7 cells for GCaMP7f and Xph20-GCaMP7f, respectively, from two independent experiments, p=0.0002 by Mann–Whitney test).

The online version of this article includes the following video, source data, and figure supplement(s) for figure 8:

**Source data 1.** Spreadsheet with the synaptic enrichment data (*Figure 8d*).

**Figure supplement 1.** Engineered calcium reporters expression.

**Figure 8—video 1.** Spontaneous responses of GCaMP6f.
https://elifesciences.org/articles/69620/figures#fig8video1

**Figure 8—video 2.** Xph15-GCaMP6f expressed in hippocampal Banker cultures and recorded at 50 Hz.
https://elifesciences.org/articles/69620/figures#fig8video2

the dendrite (*Figure 8—figure supplement 1*). Altogether, these results demonstrate that both the Xph15 and Xph20 binding modules can also be exploited to target gene-encoded module other than fluorescent proteins to excitatory synapses. In the case of the GCaMP reporter series, we also validate its compatibility with the gene regulation system in order to achieve a clear synaptic enrichment of the probe.

## Discussion

With the objective to develop imaging probes to monitor endogenous PSD-95, we have exploited a series of evolved recombinant binders of PSD-95 tandem PDZ domains derived from the $^{10}$FN3 domain. Taking advantage of both their unique paralog-specific recognition properties and their respective epitopes all situated on regions distant from the PDZ domain-binding groove, we have first confirmed that binding to their target was not detectably affecting the PDZ domains nor the full-length protein function. Their use as ReMoRA endogenous PSD-95 probes in the form of fusions to fluorescent proteins was then evaluated in comparison to both antibodies and a similar – but not specific – monobody. The tools were next further engineered to adapt them for SRI applications. We demonstrated that the resulting probes could be exploited with STED, PALM, and DNA-PAINT techniques to provide nanoscale mapping as well as dynamics information on endogenous PSD-95. Finally, we also showed that the binders can be employed to enrich active protein-based modules, such as calcium fluorescent reporters, at the excitatory postsynapse.

The three monobodies we considered in this study were all selected primarily based on their capacity to discriminate PSD-95 from other strongly homologous paralogs (PSD-93, SAP97, and SAP102). As shown before, this remarkable specificity results from their ability to recognize epitopes situated in regions distant from the targeted PDZ domain-binding groove. Indeed, while the main site of interaction of the PDZ domains with their native protein partners is conserved across paralogs, their

sequences are not strictly identical outside of these regions. Consequentially, binding of the PSD-95-specific monobodies does not obstruct the PDZ domain-binding grooves. We show here, however, that while Xph15 and Xph20, which bind exclusively to the first PDZ domain, do not detectably affect the domain nor the full-length protein function, the situation is slightly different for Xph18. Indeed, this evolved $^{10}$FN3 domain presents an epitope that encompasses regions on both PDZ domains 1 and 2. As a result, binding of Xph18 locks the two domains, otherwise free to rotate around a short linker, in a specific conformation. This conformational constraint was only detectably detrimental to the interactions of synthetic divalent PDZ domain ligands. We therefore excluded this binder from imaging applications to avoid potential impact on PSD-95 activity, even if its expression does not seem to affect AMPAR stabilization at synapses nor synaptic currents.

As previously reported, the Xph15 and Xph20 share very similar epitopes that are not known to engage in any reported PPI with neuronal partners. Importantly, these epitopes are conserved in a number of species (e.g., rodents, human, etc.) and are not subject to post-translational modifications. These features therefore guarantee a large spectrum of applications. Furthermore, we note that in the context of intrabody-like approaches, the $^{10}$FN3 scaffold, from which the binders are derived, is devoid of internal disulfide bonds, typically found, for instance, in antibody fragments, and thereby alleviating the risk of susceptibility to the intracellular reducing environment. Despite the differences in affinities and binding kinetics of Xph15 and Xph20, both allowed an efficient and specific labeling of PSD-95 independently of the associated reporter group (eGFP, mNeonGreen, SNAP-tag, mEos3.2, GCaMP). Xph20, as the tightest binder, should therefore be preferred for most applications. However, the faster binding kinetics of Xph15 can also be exploited to favor rapid renewal of the probes in live conditions, a decisive advantage when photobleaching prevents time-based experiments.

With the growing access and interest for intrabodies or their synthetic recombinant equivalents, there is a need to develop strategies to adapt the expression level of the probe to its target, in particular in the case of imaging applications. We have opted here for a regulation system developed by the group of Don Arnold for a similar application (*Gross et al., 2013*). It relies on the use of the excess (unbound) pool of probes to turn off further recombinant gene expression. In other words, the system is efficient if, on the one hand, the targeted protein is not nuclear and, on the other hand, the affinity of the evolved binder for the target is superior to the one of the appended zinc finger for its binding motif incorporated into the expression plasmid. Neuronal proteins that are located on cellular processes (dendrites or axons) are perfectly adapted for this strategy as the inevitable accumulation of fluorescent probes in the nucleus is not problematic for imaging purposes. We have observed here that the regulation system was functional for evolved binding modules with affinities in the 1–0.1 µM range in combination with a highly expressed target such as PSD-95. Indeed, for all probes and imaging techniques the expression profile was consistent with what would be expected from a directly labeled PSD-95 as confirmed by the strong colocalizations observed for the intrabodies and anti-PSD-95 antibody staining, the strong correlation obtained with FP-fused PSD-95 as well as by an estimation of synaptic copy number by PALM consistent with the literature. Furthermore, spt-PALM analysis on the millisecond timescale revealed a major population of probes in a mildly diffusive state, as would be expected from PSD-95-bound reporters.

We note, however, that while most of the reporter cargos we tried were compatible with this approach, the specific use of mScarlet-I and HaloTag resulted in the failure of the regulation system for reasons that are still unclear. The group that developed the regulation system has demonstrated that two orthogonal zinc finger systems could be used in concert (*Gross et al., 2013*). Alternative methods to regulate the effective expression levels of the probe in tune with the one of its molecular targets would also be highly valuable for multiplexing applications as well as for systems (target, binder or cargo) outside of the optimal conditions mentioned above. Developing probes that undergo fast degradation unless bound to their target constitutes an interesting alternative that has been successfully used for the nanobody scaffold (*Gerdes et al., 2020*; *Keller et al., 2018*; *Tang et al., 2016*). Another strategy for imaging applications would consist in conditioning the resulting fluorescence rather than the probe stability to its target binding by the development of fluorogenic probes (*Wongso et al., 2017*).

We have demonstrated here that the probes could be adapted to comply with a number of fluorescence-based imaging techniques. Besides the advantages of the system to monitor endogenous PSD-95 in live or fixed conditions with standard imaging procedures, SRI approaches can also

be readily accessed with adequate probe engineering. Live imaging and protein dynamics investigations can be performed by exploiting STED or spt-PALM techniques. In the case of live STED, hour timescale studies will benefit from the straightforward use of most fluorescent protein fusions, whereas for studies that require a faster temporal resolution (minute timescale), coupling of brighter and more photorobust organic dyes can be achieved by the use of the SNAP-Tag. Precise nanoscale mapping of the protein target is accessed in fixed conditions either by STED with most probes, by PALM with photoswitchable fluorescent proteins such as mEos3.2, or by DNA-PAINT with a SNAP-Tag fusion as an anchoring point for the docking DNA strand. This large variety of imaging techniques applied to endogenous proteins highlights the potential of the labeling strategy compared to more conventional labeling schemes using either antibodies or direct genetic fusions. Importantly, we note that the PSD-95 labeling observed with the different microscopy modalities here was reproducible and largely independent of the method used despite the slight differences associated with technical aspects (signal to noise, acquisition procedure, nature of the dye and its associated effective labeling efficiency, etc.). The strategy can be easily implemented to other imaging techniques, and, for instance, STORM imaging could be achieved using either the SNAP-tag or the eGFP-based probes with respectively a BG or anti-GFP nanobody functionalized with dyes possessing photoswitching properties such as those from the Cy5 cyanine family. Importantly, given the central role of PSD-95 in synaptic function, we anticipate that the probes will open up numerous possibilities for investigations against various neuronal targets by providing straightforward solutions for the monitoring of PSD-95 in the context of multi-proteins studies.

As mentioned above, two other small recombinant PSD-95 binders have been recently reported by other groups in the context of imaging applications. One is a single-chain variable fragment (scFv), PF11, that was isolated against the palmitoylated form of PSD-95 and used as an intrabody (*Fukata et al., 2013*). While the epitope was not clearly identified, the study showed recognition by the scFv of a conformational variant of PSD-95 that implied both N-terminal palmitoylation and the C-terminal part of PSD-95. Specificity was confirmed against PSD-93, one of the closest PSD-95 paralogs that also possesses a palmitoylation site in its main isoform. The other binder is a monobody, therefore, in the same synthetic recombinant binder scaffold class as Xph15 and Xph20, isolated from a selection performed against the C-terminal domains of PSD-95 (SH3 and guanylate kinase domains) (*Gross et al., 2013*). The epitope was also not determined in this study, and the isolated monobody, PSD95. FingR, was shown to also recognize SAP97 and SAP102 paralogs but not PSD-93, a property that was exploited to investigate the role of SAP97/Dlg1 in cell polarity (*Li et al., 2018*). In both cases, affinities were not determined but the binders performed efficiently as intrabody-type probes for endogenous PSD-95. However, the absence of defined epitopes for both PF11 and PSD95.FingR does not allow to convincingly rule out possible perturbations of some of PSD-95 functions when any of the two probes is bound. PSD-95 is indeed a multidomain scaffold protein with a long list of identified partners (*Dosemeci et al., 2007*; *Won et al., 2017*; *Zhu et al., 2020*) as well as numerous post-translational regulation sites (*Vallejo et al., 2017*), which complicate evaluation of the impact resulting from a recombinant binder interaction. In addition, recent studies support the idea that the protein should not be viewed as a passive scaffolding element of the synapse but rather as an active actor with a capacity to respond to partners binding (*Rademacher et al., 2019*; *Zeng et al., 2018*). In this context, we note that the results we obtained with Xph18 illustrate the difficulty to establish with certainty whether a recombinant binder may impact the activity of its target even when the epitope is known. Indeed, while we could demonstrate that this particular monobody had a clear impact on PSD-95 conformation suggesting a plausible modification of its behavior in its native environment, we did not observe detectable perturbation of PSD-95 basic functions in basal conditions.

In comparison to PF11 and FingR.PSD-95, our study shows that Xph15 and Xph20 constitute valuable complementary molecular tools for standard imaging applications based on their unique specificity profile. They recognize both palmitoylated and non-palmitoylated PSD-95 and can discriminate PSD-95 vs its paralogs. Importantly, they present the net advantage of being characterized with respect to the identity of their respective epitope. While this was critical to clarify the molecular origin of the binders specificity for PSD-95, it also allowed us to relevantly adapt their evaluation in order to confirm the absence of impact of the probes on the target protein function. Critically, Xph15 and Xph20 remarkable specificity, as well as their binding properties, has allowed us to engineer the binders as SRI probes to investigate endogenous PSD-95.

Besides the use of evolved recombinant binders as a strategy to label endogenous PSD-95 in live conditions, a number of genetic approaches have been reported. They all rely on gene-editing methods and are typically used to generate PSD-95 fluorescent protein (*Broadhead et al., 2016*; *Fortin et al., 2014*; *Willems et al., 2020*; *Zhu et al., 2020*) or engineered self-labeling enzyme fusions (*Masch et al., 2018*). Comparatively, their main advantages over expressed exogenous probes are the ideal stoichiometric labeling (one fluorophore per target protein, to be tempered by the notion of effective labeling efficiency of the fluorophore; *Thevathasan et al., 2019*) together, for the knock-in approaches, with the possibility to achieve global labeling.

In contrast, the ReMoRA or intrabody-based approaches benefit from their ease of implementation by relying on standard cell biology techniques for the genetically encoded probe delivery (transfection, electroporation, or virus-mediated delivery). Indeed, gene-editing methods are still not accessible in routine use to most laboratories and are also not amenable to downstream adaptation to various imaging techniques, the possibilities being imposed by the initial choice of the fluorescent module. The modular design of the recombinant binder-based probes provides in turn a system more adapted to engineering and optimization (binding module, fluorescent system, promoter, etc.). Furthermore, we note that the rapid renewal of the probe obtained with the fast kinetics binders can be advantageous for imaging purposes over genetic modification of PSD-95 as its turnover is slow in basal conditions.

Finally, if we consider altogether the properties of the tools presented here and compare them to existing probes based on other recombinant PSD-95 binders (PF11 and PSD95.FingR) or to other labeling approaches (e.g., gene editing), we can identify a number of biological applications that cannot be currently achieved and that would now become accessible. Indeed, Xph15- and Xph20-derived probes currently constitute the only genetically encoded systems that allow the strict specific detection of endogenous PSD-95 for live and advanced imaging applications with, in addition, a comprehensive description of their target binding properties (epitopes, affinities and kinetics, absence detectable interference, etc.). The latter aspect is critical for ensuring that the tools can be exploited without modifying PSD-95 regulation and function and in turn not lead to artifactual results regardless of whether PSD-95 is the prime object of the investigation or just used as an excitatory postsynapse marker. This also constitutes unvaluable information for further engineering and molecular tool development. Specificity is the other salient feature of our tools, and along these lines, their full potential will be unlocked with their exploitation for comparative studies of the PSD-95 paralogs. As mentioned above, our tools provide an unprecedented access to endogenous PSD-95-specific labeling strategies that can be used in combination with complementary strategies to label PSD-93, SAP102, or SAP97, either existing ones with their respective limitations (antibodies, FP-fusion, etc.) or, ideally, with new sets of intrabodies that would here be specific for each of the other paralogs. A multiplexed ReMoRA approach to label the PSD-95 paralogs would open numerous possibilities for investigating the respective role of each of these major neuronal actors notably by providing access to advance imaging approaches with minimal interference on the targeted endogenous proteins.

In conclusion, we provide here a set of powerful probes for targeting PSD-95 with main applications for endogenous protein imaging as well as synaptic enrichment of active protein modules such as activity reporters. In comparison to other similar existing tools, the evolved monobodies described here constitute to this day the only binding modules displaying a strict specificity for PSD-95 regardless of its post-translational modification state. The molecular understanding of their mode of binding comforts our results, indicating undetectable perturbation of PSD-95 function. The probes presented here, which benefit from the simplicity of use of the ReMoRA design, provide direct access to different SRI techniques. We anticipate that beyond the direct benefit for nanoscale mapping and (single molecule) dynamics investigations of endogenous PSD-95, these probes will turn invaluable for investigations that require the implementation of multiplexing imaging strategies.

## Materials and methods

**Key resources table**

| Reagent type (species) or resource | Designation | Source or reference | Identifiers | Additional information |
|---|---|---|---|---|
| Strain, strain background (*Escherichia coli*) | BL21 CodonPlus (DE3)-RIPL | Agilent | Cat# 230280 | |
| Strain, strain background (*E. coli*) | T7 Express lysY | New England Biolabs | Cat# C3010I | |
| Cell line (simian) | COS-7 | ECACC-87021302 | | |
| Antibody | Anti-GluA2 ATTO-647N (mouse monoclonal) | Gift from Eric Gouaux, coupled in lab. | PMID:23926273 | 30 ng/ml |
| Antibody | Anti-PSD-95 (mouse monoclonal) | Thermo Fisher | Cat# MA1-046 | 1:1000 |
| Antibody | Goat anti-mouse Alexa Fluor 568 (goat polyclonal) | Thermo Fisher | Cat# A-11031 | 1:1000 |
| Antibody | Anti-MAP2 (chicken polyclonal) | Synaptic Systems | Cat# 188 006 | 1:2000 |
| Antibody | Goat anti-chicken Alexa Fluor 594 (goat polyclonal) | Thermo Fisher | Cat# A-11042 | 1:800 |
| Recombinant DNA reagent | Numerous | See *Supplementary file 2* | | https://www.addgene.org/Matthieu_Sainlos/ |
| Sequence-based reagent | Numerous | See *Supplementary file 3* | | |
| Peptide, recombinant protein | PSD-95-12 [61-249] | *Rimbault et al., 2019* | PMID:31586061 | |
| Peptide, recombinant protein | Xph20 | *Rimbault et al., 2019* | PMID:31586061 | https://www.addgene.org/Matthieu_Sainlos/ |
| Peptide, recombinant protein | Xph18 | *Rimbault et al., 2019* | PMID:31586061 | https://www.addgene.org/Matthieu_Sainlos/ |
| Peptide, recombinant protein | Xph15 | *Rimbault et al., 2019* | PMID:31586061 | https://www.addgene.org/Matthieu_Sainlos/ |
| Peptide, recombinant protein | Xph0 | *Rimbault et al., 2019* | PMID:31586061 | https://www.addgene.org/Matthieu_Sainlos/ |
| Peptide, recombinant protein | Stg15 (Ac-YSLHANTANRRTTPV) | *Rimbault et al., 2019* | PMID:31586061 | |
| Peptide, recombinant protein | FITC-Stg15 (FITC-PEG-YSLHANTANRRTTPV) | *Rimbault et al., 2019* | PMID:31586061 | |
| Peptide, recombinant protein | [Stg15]2 | *Rimbault et al., 2019* | PMID:31586061 | |
| Peptide, recombinant protein | FITC-[Stg15]2 | *Rimbault et al., 2019* | PMID:31586061 | |
| Commercial assay or kit | X-treme GENE HP DNA transfection reagent | Roche | | |
| Commercial assay or kit | Effectene Kit | QIAGEN | | |
| Chemical compound, drug | SNAP-Cell 647-SiR (BG-SiR) | New England Biolabs | Cat# S9102S | |
| Software, algorithm | PyMOL | Warren DeLano | RRID:SCR_000305 | |
| Software, algorithm | Fiji | PMID:22743772 | RRID:SCR_002285 | |
| Software, algorithm | Adobe Illustrator | Adobe Systems | RRID:SCR_010279 | |
| Software, algorithm | Prism 7.04, 8 | GraphPad | RRID:SCR_002798 | |
| Software, algorithm | TopSpin v4.0 | Bruker | RRID:SCR_014227 | |
| Software, algorithm | MetaMorph v7.8.10.0 | Molecular Devices | RRID:SCR_002368 | |
| Software, algorithm | LI-FLIM v1.2.12 | Lambert Instruments | | |
| Software, algorithm | POLARstar Omega v5.11 | BMG Labtech | | |
| Software, algorithm | NMRPipe v8.6 | *Delaglio et al., 1995* | PMID:8520220 | |

| Reagent type (species) or resource | Designation | Source or reference | Identifiers | Additional information |
|---|---|---|---|---|
| Software, algorithm | Sparky v3.113 | D. Goddard and D. G. Kneller, SPARKY 3, University of California, San Francisco | RRID:SCR_014228 | |
| Software, algorithm | POLARstar MARS data analysis software v3.20 | BMG Labtech | | |
| Software, algorithm | Patchmaster | Heka Elektronik | | |
| Software, algorithm | SR-Tesseler | *Levet et al., 2015* | PMID:26344046 | |
| Software, algorithm | IJ-Macro_FRAP-MM | https://github.com/fabricecordelieres/IJ-Macro_FRAP-MM | | |
| Software, algorithm | IJ-Plugin_Metamorph-Companion | https://github.com/fabricecordelieres/IJ-Plugin_Metamorph-Companion | | |
| Software, algorithm | PICASSO | *Schnitzbauer et al., 2017* | PMID:28518172 | |
| Other | SNAPligand-modified DNA oligomer | *Schnitzbauer et al., 2017* | PMID:28518172 | 5' BG-TTATACATCTA 3' |
| Other | Cy3b-labeled DNA imager strands | *Schnitzbauer et al., 2017* | PMID:28518172 | 5' CTAGATGTAT-Cy3b 3' |

## Plasmid construction

The plasmids generated and the primers used in this study are listed in *Supplementary files 2 and 3*, respectively. Plasmids for protein production were described previously (*Rimbault et al., 2019*; *Sainlos et al., 2009*). Briefly, for bacterial expression, the first two PDZ domains of PSD-95 were subcloned into pET-NO to produce an N-terminal fusion with an octa-His tag and a TEV protease cleavage site. The Xph clones were subcloned into the pIGc vector to generate C-terminal fusions with a deca-His tag. For FRET experiments, PSD-95-eGFP and stargazin-mCherry were described previously (*Sainlos et al., 2011*). Plasmids for soluble Xph clone expression were obtained by replacing the eGFP-CCR5 ZF-KRAB(A) fragment from the corresponding pCAG vector (gift from Don Arnold, USC, Addgene #46295; *Gross et al., 2013*) by an octa-His and HA tags using BglII and BsrGI restriction sites. Plasmid for soluble PSD-95 PDZ domain 2 expression was obtained by first replacing eGFP into pEGFP-N1 by mIRFP via BamHI and BsrGI restriction sites (gift from M Davidson, Florida State University, and X Shu, UCSF, Addgene #54620; *Yu et al., 2015*) and then subcloning the PDZ domain using HindIII and BamHI restriction sites. The PDZ domain-based FRET reporter was obtained as described previously (*Rimbault et al., 2019*) but here without mutation of the first domain. The plasmid for expression of soluble Xph15 and 18 with a miRFP670 nuclear reporter was generated as described for the one with Xph20 (*Rimbault et al., 2019*).

For imaging, Xph15, Xph18, and Xph20 were subcloned into pCAG_PSD95.FingR-eGFP-CCR5TC (gift from Don Arnold, USC, Addgene #46295; *Gross et al., 2013*) using KpnI and BglII restriction sites. Other fluorescent modules, mRuby2 (gift from Michael Lin, Addgene #40260; *Lam et al., 2012*), mScarlet-I (gift from Dorus Gadella, Addgene #98821; *Bindels et al., 2017*), mEos3.2 (gift from Michael Davidson and Tao Xu, Addgene 54525; *Zhang et al., 2012*), mNeonGreen (obtained by gene synthesis, Eurofins; *Shaner et al., 2013*), HaloTag (Promega, Cat# G7971), and SNAPf (New England Biolabs, Cat# N9183S) were next inserted in place of eGFP in the corresponding vector using BglII and NheI sites after an initial modification of the source vectors to introduce an NheI site between the fluorescent module and CCR5 ZF. GCaMP6f (*Chen et al., 2013*) and GCaMP7f (*Dana et al., 2019*) expressing plasmids were gifts from Douglas Kim & GENIE Project (Addgene #40755 and #104483, respectively). Xph15 was subcloned as an N-terminal fusion to GCaMP6f by using BglII and SalI restriction sites and GCaMP7f was subcloned C-terminally to Xph20 into pCAG_Xph20-eGFP-CCR5TC using BglII and either MluI or NheI sites for removal or conservation of the eGFP-CCR5 ZF-KRAB(A) fragment, respectively.

AAV-expressing vector containing Xph15, Xph18, or Xph20 was subcloned into AAV-Syn-PSD95.FingR-eGFP-CCR5TC (gift from Xue Han, Addgene plasmid# 125693; *Bensussen et al., 2020*) by

replacing the PSD95.FingR coding sequence using the NheI and SphI restriction sites. The pAAV Syn-eGFP construct (a gift from Edward Boyden, Addgene plasmid# 58867; *Boyden et al., 2005*) carries only the soluble eGFP sequence without the CCR5TC sequences and ZF sites upstream of the synapsin promoter.

PSD-95 shRNA_mCherry construct was obtained by replacing the eGFP by mCherry using AgeI and BsRGI restriction sites in the lentiviral vector FH95pUGW (B3) (gift from Robert Malenka and Oliver Schluter and Weifeng Xu, Addgene #74012; *Schlüter et al., 2006*). The shRNA-negative control construct (scramble shRNA_mCherry) was derived from the previous construct by inserting the H1 promoter and a non-effective mammalian scramble sequence (source: pSUPER.Mamm-x, Oligoengine, Cat# vec-neg-0002) was PCR amplified and inserted using the EcoRI and PacI restriction sites.

PSD-95-mScarlet-I was obtained by replacing eGFP into PSD-95-eGFP C-terminal fusion construct (*Sainlos et al., 2011*) via AgeI and BsRGI restriction sites in a pEGFP-N1 clontech vector backbone.

## Protein production

Proteins were expressed and purified as described previously (*Rimbault et al., 2019*). Briefly, His-tagged proteins were either produced in *Escherichia coli* BL21 CodonPlus (DE3)-RIPL competent cells (Agilent, 230280) using auto-induction protocols (*Studier, 2005*) at 16°C for 20 hr or in BL21 pLysY (New England Biolabs, C3010I) for isotopically labeled proteins with IPTG induction for 16 hr at 20°C. Proteins were first isolated by IMAC using Ni-charged resins then further purified by size-exclusion chromatography (SEC). An intermediate step of affinity tag removal by incubation with the TEV protease was added before the SEC step for isotopically labeled proteins. The recovered proteins were concentrated, aliquoted, and flash-frozen with liquid nitrogen for conservation at –80°C.

## Peptide synthesis

Peptides were synthesized as described previously (*Rimbault et al., 2019*). Briefly, amino acids were assembled at 0.05 mmol scale by automated solid-phase peptide synthesis on a CEM µwaves Liberty-1 synthesizer (Saclay, France) following standard coupling protocols. The divalent ligand $[Stg_{15}]_2$ was obtained by using copper-catalyzed click chemistry on resin harboring a mix of sequences functionalized by azide and alkyne groups as described previously (*Sainlos et al., 2011*). Briefly, a 7:3.5 mixture of Fmoc-Lys($N_3$)-OH and pentynoic acid was manually coupled to the deprotected N-terminal amino group of elongated peptides on resin followed by copper(I)-catalyzed azide-alkyne cycloaddition in DMF/4-methylpiperidine (8:2) with CuI (5 eq), ascorbic acid (10 eq), and aminoguanidine (10 eq). N-free peptide resins were derivatized with acetyl groups or further elongated with a PEG linker (Fmoc-TTDS-OH, 19 atoms, Iris Biotech, FAA1568) and fluorescein isothiocyanate. Peptides were purified by RP-HPLC with a semi-preparative column (YMC $C_{18}$, ODS-A 5/120, 250 × 20 mm) and characterized by analytical RP-HPLC and MALDI-TOF. Peptides were lyophilized and stored at –80°C until usage.

## NMR spectroscopy

NMR spectra were recorded at 298 K using a Bruker Avance III 700 MHz spectrometer equipped with a triple resonance gradient standard probe. Topspin version 4.1 (Bruker BioSpin) was used for data collection. Spectra processing used NMRPipe (*Delaglio et al., 1995*) with analysis by using Sparky 3 (T. D. Goddard and D. G. Kneller, University of California). Titration of 200 µM $^{15}$N-labeled PSD-95 PDZ1-PDZ2 in PBS with a stock solution of 10 mM Stargazin C-terminal peptide (Ac-YSLHANTANR-RTTPV) was followed by 1D $^1$H and 2D $^{15}$N-HSQC spectra. Titration points include 0, 40, 80, 120, 160, 200, 240, 280, 320, 360, 400, and 440 µM peptide, corresponding to 0.2, 0.4, 0.6, 0.8, 1, 1.2, 1.4, 1.6, 1.8, 2, and 2.2 molar equivalents peptide:protein. The titration was repeated by using pre-assembled 1:1 complexes of 200 µM $^{15}$N-labeled PSD-95 PDZ1-PDZ2 with a slight molar excess (240 µM) of natural abundance Xph15, Xph18, or Xph20. Amide $^1$H,$^{15}$N chemical shift assignments of unbound and bound [$^{15}$N]PSD-95-12 were previously reported (*Rimbault et al., 2019*).

## Fluorescence polarization assay

For direct titrations, the fluorescein-labeled stargazin peptide (10 nM) was titrated against a range of increasing concentrations of the different recombinant PDZ domains in a 100 µl final volume. Fluorescence polarization was measured in millipolarization units (mP) at an excitation wavelength of

485 ± 5 nm and an emission wavelength of 520 ± 5 nm using a POLARstar Omega (BMG Labtech) microplate reader. Titrations were conducted at least in duplicate and measured twice. To determine the corresponding affinities (apparent $K_D$), curves were fitted using a nonlinear regression fit formula (*Chang et al., 2011*) with GraphPad Prism v7.04 after normalizing the values of each protein series between the initial unbound and the saturating states.

For competitive titrations, experiments were designed such that the starting polarization value represents 75% of the maximal shift of the direct titrations. For the divalent stargazin ligand, PSD-95-12 was used at a concentration of 90 nM. Tandem PDZ domains, bound to the fluorescein-labeled stargazin divalent peptide (10 nM), were titrated against a range of increasing concentrations of acetylated stargazin divalent ligand in a 100 µl final volume in the presence of 5 µM of Xph clones. For the monovalent stargazin ligand, PSD-95-12 (at a concentration of 20 µM), bound to the fluorescein-labeled stargazin monovalent peptide (50 nM), was titrated against a range of increasing concentrations of stargazin peptides in a 100 µl final volume in the presence of 20 µM of Xph18. Titrations were conducted as above at least in duplicate and measured three times. To determine the corresponding inhibition constant ($K_I$), curves were fitted using a competition formula (*Pazos et al., 2011*) with GraphPad Prism v7.04 after normalizing the values of each protein series between the initial unbound and the saturating states.

## FRET/FLIM assays

FRET/FLIM assays were performed as described previously (*Rimbault et al., 2019*). Briefly, COS-7 cells (ECACC-87021302) in DMEM medium supplemented with GlutaMAX and 10% FBS were transfected using a 2:1 ratio X-treme GENE HP DNA transfection reagent (Roche) per µg of plasmid DNA with a total of 0.5 µg DNA per well. Experiments were performed after 24 hr of expression. Coverslips were transferred into a ludin chamber filled with 1 ml fresh Tyrode's buffer (20 mM glucose, 20 mM HEPES, 120 mM NaCl, 3.5 mM KCl, 2 mM $MgCl_2$, 2 mM $CaCl_2$, pH 7.4, osmolarity around 300 mOsm/kg and pre-equilibrated in a $CO_2$ incubator at 37°C).

Experiments with full-length PSD-95 were performed using the time domain analysis (TCSPC) method with a Leica DMR TCS SP2 AOBS on an inverted stand (Leica Microsystems, Mannheim, Germany). The pulsed light source was a tunable Ti:Sapphire laser (Chameleon, Coherent Laser Group, Santa Clara, CA) used at 900 nm and 80 MHz, providing a 13 ns temporal window for lifetime measurements. The system was equipped with the TCSPC from Becker and Hickl (Berlin, Germany), and fluorescence decay curves were obtained using single-spot mode of SPCM software (Becker and Hickl).

Experiments with the PSD-95-12-derived FRET reporter system were performed using the frequency domain analysis (LIFA) method and Leica DMI6000 (Leica Microsystems, Wetzlar, Germany) equipped with a confocal Scanner Unit CSU-X1 (Yokogawa Electric Corporation, Tokyo, Japan). The FLIM measurements were done with the LIFA fluorescence lifetime attachment (Lambert Instrument, Roden, the Netherlands), and images were analyzed with the manufacturer's software LI-FLIM software.

Lifetimes were referenced to a 1 µM solution of fluorescein in Tris–HCl (pH 10) or a solution of erythrosin B (1 mg/ml) that was set at 4.00 ns lifetime (for fluorescein) or 0.086 ns (for erythrosin B). For competition experiments, only cells presenting a high level of expression of the competitor or control as measured by mIRFP670 fluorescence were taken into consideration.

## Cell culture

All experiments were performed on E18 rat dissociated hippocampal culture except electrophysiology experiments that have been performed in mice P1 hippocampal culture. Banker culture from rat hipppocampal E18 culture neurons were prepared using a previously described protocol (*Penn et al., 2017*) with the following modifications: neuron cultures were maintained in Neurobasal medium (Cat# 12348017, Thermo Fisher Scientific) supplemented with 2 mM L-glutamine (Thermo Fisher Scientific, Cat# 25030-024) and SM1 Neuronal Supplement (Cat# 05711, STEMCELL Technologies).

The interactomics and knock-down-shRNA interference experiments were performed on rat E18 dissociated culture maintained in Neurobasal Plus medium supplemented with 0.5 mM GlutaMAX and 1× B-27 Plus supplement (Thermo Fisher Scientific).

## Gene delivery

For electrophysiology experiments, rat hippocampal neurons were transfected with Xph15 or Xph18 using Effectene kit (QIAGEN N.V., Venlo, the Netherlands) at 7–9 DIV. For immunostaining, FRAP, correlation, STED, and DNA-PAINT experiments, rat hippocampal neurons from E18 embryos were electroporated at DIV 0 before plating with 1.5 µg of DNA using Nucleofector system (Lonza) in 100 µl Single Nucleocuvette with P3 Primary Cell 4D-Nucleofector X Kit and HV hippocampal neuron program. For DNA-PAINT and PSD-95 knock-down experiments, primary hippocampal neurons were transfected using a standard calcium phosphate protocol at DIV 7–8 with Xph20-SNAP or Xph20-HaloTag and a soluble eGFP (DNA-PAINT) or with PSD-95 shRNA_mCherry or scramble shRNA_mCherry and Xph15-eGFP or Xph20 eGFP (knock-down).

## Electrophysiology

### Mouse hippocampal mass-culture neuron culture and AAV infection

Primary mouse hippocampal cultures were prepared as described previously (*Xue et al., 2008*). Briefly, hippocampal neurons were prepared from postnatal day 1 mice from both sexes. Hippocampal neurons were plated on continental WT astrocyte feeder layer. The viral production was performed by the Viral Core Facility of the Charité-Universitätsmedizin Berlin. Hippocampal neurons were infected with adenoviruses (AAV2/1) at DIV 2–3 and left at 37°C and 5% $CO_2$ until the electrophysiological experiments were performed.

### Mouse hippocampal mass-culture neurons electrophysiology

Whole-cell voltage clamp experiments were performed on approximately equal numbers of mouse hippocampal mass-culture neurons from each group in parallel on the same day in vitro (11–16 DIV) at room temperature (RT) (23–24°C). Neurons were clamped at −70 mV with an Multiclamp 700B amplifier (Molecular Devices) under the control of Clampex 10.5 software (Molecular Devices). Data were acquired using an Axon Digidata 1550 Digitizer (Axon Instruments) at 10 kHz and low-pass Besser filtered at 3 kHz. Borosilicate glass pipettes with 2–5 MΩ resistance were pulled with a micropipette puller device (Sutter Instruments). Only cells with series resistances below 12 MΩ after break-in were analyzed. The pipettes were filled with intracellular solution (ICS) containing the following (in mM): 136 KCl, 17.8 HEPES, 1 EGTA, 4.6 $MgCl_2$, 4 $Na_2ATP$, 0.3 $Na_2GTP$, 12 creatine phosphate, and 50 U/ml phosphocreatine kinase (~300 mOsm, pH 7.4). The standard extracellular solution (ECS) contained (in mM) 140 NaCl; 2.4 KCl, 10 HEPES, 10 glucose, 2 $CaCl_2$, 4 $MgCl_2$, 300 mOsm; pH 7.4. mEPSCs were recorded in standard external solution with 0.5 µM tetrodotoxin and 15 µM gabazine. Data were analyzed offline using AxoGraph X (AxoGraph Scientific). To detect mEPSC events, traces were digitally filtered at 1 kHz offline and events were automatically selected with a scaled-template algorithm (*Clements and Bekkers, 1997*) in AxoGraph X. The template function is a double exponential with a scalable amplitude, a rise time constant of 0.5 ms, a decay time constant of 4 ms, a baseline of 5 ms, and a template length of 10 ms. mEPSC frequencies and amplitudes were determined 2 min after establishing whole-cell configuration and for a period of 120 s time window. Statistic significances were tested using the nonparametric Kruskal–Wallis test followed by Dunn's post hoc test when at least one group showed a nonparametric distribution. When all groups passed a normality test, we applied a one-way ANOVA followed by Turkey's post hoc test.

## Electrophysiology in rat primary hippocampal neurons

Whole-cell patch clamp recordings were performed on Banker cultures of hippocampal neurons (13–17 DIV) expressing Xph15 or Xph18 fused to eGFP. The experiments were carried out at RT in an ECS containing the following (in mM): 110 NaCl, 5.4 KCl, 10 HEPES, 10 glucose, 1.8 $CaCl_2$, 0.8 $MgCl_2$ (Sigma-Aldrich, St-Louis); 250 mOsm; pH 7.4. To block voltage-gated sodium channels, 1 µM tetrodotoxin (TTX; Tocris Bioscience, Bristol, UK) was added to the ECS. ICS contained the following (in mM): 110 K-gluconate, 1.1 EGTA, 10 HEPES, 3 $Na_2ATP$, 0.3 $Na_2GTP$, 0.1 $CaCl_2$, 5 $MgCl_2$ (Sigma-Aldrich); 240 mOsm; pH 7.2. Patch pipettes were pulled using a horizontal puller (P-97, Sutter Instrument) from borosilicate capillaries (GB150F-8P, Science Products GmbH) to resistances of 3–5 MΩ when filled with ICS. All recordings were performed using an EPC10 patch clamp amplifier operated with Patchmaster software (HEKA Elektronik). Data was acquired at 10 kHz and filtered at 3 kHz. Membrane capacitance

was monitored frequently throughout the experiments, and only cells with a series resistance <10 MΩ were analyzed.

Data were collected and stored on computer for offline analysis using a software developed in-house (Detection Mini) to detect miniature synaptic events using a variable threshold. The amplitude and frequency of mEPSCs were obtained for a minimum of 500 events.

Statistical values are given as mean ± SEM. Statistical significances were performed using GraphPad Prism software (San Diego, CA). Normally distributed datasets were tested by Student's unpaired t-test for two independent groups.

## uPAINT

uPAINT was performed as previously reported (*Giannone et al., 2010*) on dissociated neurons expressing Xph15, Xph18, or Xph20 fused to eGFP. Experiments took place at 13–16 DIV. Cells were imaged at 37°C in an open chamber (Ludin chamber, Life Imaging Services, Switzerland) filled with 1 ml of Tyrode's solution (in mM): 10 HEPES, 5 KCl, 100 NaCl, 2 $MgCl_2$, 2 $CaCl_2$, 15 glucose (pH 7.4). The chamber was mounted on an inverted microscope (Nikon Ti-Eclipse, Japan) equipped with a high ×100 objective (1.49 NA), a TIRF device, and an EMCCD camera (Evolve camera; Roper Scientific, Princeton Instruments, Trenton, NJ). Dendritic regions of interest (ROIs) were selected based on eGFP signal. To track endogenous GluA2-containing AMPAR, an anti-GluA2 antibody given by E. Gouaux (Portland, OR) coupled to ATTO-647N (Atto-Tec, Siegen, Germany) was used. Stochastic labeling of the targeted protein by dye-coupled antibodies allowed the recording of thousands of trajectories lasting longer than 1 s. Recordings were made at 50 Hz using MetaMorph software (Molecular Devices, USA), and analysis was performed with a homemade software developed under MetaMorph and kindly provided by J.B. Sibarita (Interdisciplinary Institute for Neuroscience).

## Interactomics

For proteomic experiments, mixed E18 rat hippocampal cultures were plated at 600k/well on 6-well plates. At 3 DIV, neurons were transduced using AAV2/9 containing Xph20-eGFP or soluble eGFP as a control (MOI 75k). Cultures were fed at 3 DIV and 13 DIV. At 17 DIV, cells were lysed on ice with 100 µl per well of lysis buffer (125 mM NaCl, 25 mM HEPES, 1% NP40, 1× protease inhibitor cocktail [Calbiochem]). Cell lysates were collected, homogenized, and centrifuged at 12,500 rpm for 10 min. Protein concentration of each lysate was quantified using BCA reagent (Thermo Fisher Scientific). For PSD-95 immunoprecipitation, 1 mg of protein per condition was incubated 60 min at 4°C with 80 µl of Dynabeads protein-G (Invitrogen, Cat# 10004D) pre-coated with 30 µg of mouse anti-PSD-95 antibody (Sigma-Aldrich, Cat# MAB1596) during 20 min at RT. The immunoprecipitations were washed three times with PBS-Tween20 0.02% buffer and eluted in 80 µl SDS-PAGE loading buffer. 65 µl was used for proteomic analysis.

Protein samples were solubilized in Laemmli buffer, and samples were deposited in triplicate onto SDS-PAGE gel. After colloidal blue staining, each lane was cut out from the gel and was subsequently cut in 1 mm × 1 mm gel pieces. Gel pieces were destained in 25 mM ammonium bicarbonate 50% ACN (acetonitrile), rinsed twice in ultrapure water, and shrunk in ACN for 10 min. After ACN removal, gel pieces were dried at RT, covered with the trypsin solution (10 ng/µl in 50 mM $NH_4HCO_3$), rehydrated at 4°C for 10 min, and finally incubated overnight at 37°C. Spots were then incubated for 15 min in 50 mM $NH_4HCO_3$ at RT with rotary shaking. The supernatant was collected, and an $H_2O$/ACN/HCOOH (47.5:47.5:5) extraction solution was added onto gel slices for 15 min. The extraction step was repeated twice. Supernatants were pooled and dried in a vacuum centrifuge. Digests were finally solubilized in 0.1% HCOOH.

## nLC-MS/MS analysis and label-free quantitative data analysis

Peptide mixture was analyzed on a Ultimate 3000 nanoLC system (Dionex, Amsterdam, the Netherlands) coupled to an Electrospray Orbitrap Fusion Lumos Tribrid Mass Spectrometer (Thermo Fisher Scientific, San Jose). Then, 10 µl of peptide digests were loaded onto a 300-µm-inner diameter × 5 mm $C_{18}$ PepMap trap column (LC Packings) at a flow rate of 10 µl/min. The peptides were eluted from the trap column onto an analytical 75 mm id × 50 cm $C_{18}$ Pep-Map column (LC Packings) with a 4–40% linear gradient of solvent B in 91 min (solvent A was 0.1% formic acid and solvent B was 0.1% formic acid in 80% ACN). The separation flow rate was set at 300 nl/min. The mass spectrometer operated

in positive ion mode at a 1.9-kV needle voltage. Data were acquired using Xcalibur 4.4 software in a data-dependent mode. MS scans (*m/z* 375–1500) were recorded in the Orbitrap at a resolution of $R =$ 120,000 (@ *m/z* 200) and an AGC target of $4 \times 10^5$ ions collected within 50 ms. Dynamic exclusion was set to 30 s and top-speed fragmentation in HCD mode was performed over a 3 s cycle. MS/MS scans were collected in the Orbitrap with a resolution of 30,000 and maximum fill time of 54 ms. Only +2 to +6 charged ions were selected for fragmentation. Other settings were as follows: no sheath nor auxiliary gas flow, heated capillary temperature, 275°C; normalized HCD collision energy of 28%, isolation width of 1.6 *m/z*, AGC target of $5 \times 10^4$ and normalized AGC target of 100%. Monoisotopic precursor selection (MIPS) was set to Peptide and an intensity threshold was set to $2.5 \times 10^4$.

### Database search and results processing

Data were searched by SEQUEST through Proteome Discoverer 2.5 (Thermo Fisher Scientific Inc) against the *Rattus norvegicus* Reference Proteome Set from UniProt (29,918 entries in v2021-03) and the sequences of Xph20 and eGFP. Spectra from peptides higher than 5000 Da or lower than 350 Da were rejected. Precursor detector node was included. Search parameters were as follows: mass accuracy of the monoisotopic peptide precursor and peptide fragments was set to 10 ppm and 0.02 Da, respectively. Only b- and y-ions were considered for mass calculation. Oxidation of methionines (+16 Da), methionine loss (–131 Da), methionine loss with acetylation (–89 Da), and protein N-terminal acetylation (+42 Da) were considered as variable modifications while carbamidomethylation of cysteines (+57 Da) was considered as fixed modification. Two missed trypsin cleavages were allowed. Peptide validation was performed using Percolator algorithm (*Käll et al., 2007*) and only 'high-confidence' peptides were retained corresponding to a 1% false positive rate at peptide level. Peaks were detected and integrated using the Minora algorithm embedded in Proteome Discoverer. Proteins were quantified based on unique peptides intensities. Normalization was performed based on total human protein amount. Protein ratios are calculated from the group protein abundances. An ANOVA was calculated for each individual protein with Benjamini–Hochberg correction. Quantitative data were considered for proteins quantified by a minimum of two peptides and a p-value lower than 0.05. The list of identified PSD-95 (entry P31016) binding partners (435 entries) was taken from the Molecular INTeraction (MINT) public database (https://mint.bio.uniroma2.it/).

The mass spectrometry proteomics data have been deposited to the ProteomeXchange Consortium via the PRIDE (*Perez-Riverol et al., 2022*) partner repository with the dataset identifier PXD045002.

## Immunostaining

At 23–27 DIV, E18 rat Banker cultures expressing individual eGFP-tagged Xph or PSD95.FingR were stained with mouse monoclonal anti-PSD-95 (Thermo Fischer Scientific, Cat# MA1-046) at 1 µg/ml final concentration. Briefly, neurons on coverslips were fixed 10 min using PFA 4%, washed with PBS, permeabilized with PBS-Triton-0.1% during 5 min, and washed again. After blocking with PBS-BSA 0.5%, neurons were stained with the PSD-95 antibody and after three washes with a secondary antibody (goat anti-mouse Alexa 568, Cat# A111031) for 45 min each. Neuron coverslips were mounted on Pro-Long Gold antifade reagent (Thermo Fischer Scientific, Cat# P36934).

Images were acquired on a Leica DM5000 (Leica Microsystems) with a HCX PL APO ×63 oil NA 1.40 objective, a LED SOLA Light (Lumencor, Beaverton) as fluorescence excitation source and a Flash4.0 V2 camera (Hamamatsu Photonics, Massy, France). Image quantifications were performed using tasks automatization with MetaMorph. Following a background subtraction, the images were automatically thresholded to detect the positive objects for the recombinant binders (Xph15, Xph18, Xph20, or PSD95.FingR) and PSD-95. Enrichment was measured by the ratio between the fluorescence intensity of the positive objects for the recombinant binders and the shaft. Object colocalization was evaluated by determining ROIs around positive objects for the recombinant binders and measuring the fluorescence intensity of these regions in the channel of PSD-95.

## FRAP

Banker cultures (21–23 DIV) in coverslips expressing eGFP fusions of Xph clones or full-length PSD-95 were mounted in a Ludin chamber (Life Imaging Services) and transferred to an inverted microscope (Leica, DMI 6000B) maintained at 37°C. Fluorescence experiments were carried out in an ECS containing (in mM) 120 NaCl, 3.5 KCl, 2 MgCl$_2$, 2 CaCl$_2$, 10 D-glucose, 10 HEPES (pH 7.4, ~270 mOsm),

and transfected neurons were observed through a ×63 oil objective (Leica, HC PL APO CS2, NA 1.4). GFP fluorescence was illuminated with 491 nm laser light using a high-speed spinning disk confocal scanner unit (Yokogawa CSU22-W1) and emission was captured with a sCMOS camera (Prime 95B, Photometrics). Microscope hardware was controlled with MetaMorph (Molecular Devices, v7.1.7) and ILAS2 system (Gataca Systems, Massy, France) software.

For FRAP experiments, the following protocol was used: (1) prebleaching (20 images at 3 s interval); (2) photobleaching of the ROIs (10–15 ROIs per field of view, ROI = 10 pixels, eq. to 2.3 µm), (3) fast recovery (40 images at 0.5 s interval), and (4) long-term recovery (10 min recording at 3 s interval). For photobleaching, we used a 5 ms pulse of 488 nm laser light sufficient to reduce fluorescence by at least 50%. Experiments where fluorescence dropped more than 20% in non-bleached regions during acquisition were discarded.

FRAP experiments were analyzed using an in-house-developed macro to the ImageJ freeware (http://imagej.nih.gov/ij/). The source code is freely available from GitHub (https://github.com/fabricecordelieres/IJ-Macro_FRAP-MM; *Cordelières, 2019a*), accompanied by a documentation and an example dataset. Briefly, as part of the macro, the FRAP region for each spine was imported from the MetaMorph software to ImageJ's ROI Manager using the MetaMorph Companion plugin (https://github.com/fabricecordelieres/IJ-Plugin_Metamorph-Companion; *Cordelières, 2019b*). From the data extracted by the macro, average intensity within the ROI was collected for each timepoint ($F_t$), at first timepoint (pre-bleach, $F_{pb}$), and immediately after the bleaching ($F_0$). Simple normalization was performed as follows: $FRAP_t = F_t - F_0/F_{pb} - F_0$. The mean spine FRAP curve of each cell was subsequently fitted to a monoexponential model using GraphPad Prism software.

## Knock-down (shRNA)

Xph15 or Xph20-eGFP-CCR5TC were co-expressed in E18 rat hippocampal cultures with either PSD-95 shRNA_mCherry or scramble shRNA_mcherry (2 µg of each DNA per condition). After 4–5 d (8 DIV to 13 DIV) of expression, banker neuronal cultures were fixed, permeabilized with 0.1 Triton-X-100, and immunolabeled with a mouse monoclonal anti-PSD-95 (Thermo Fisher Scientific, Cat# MA1-046) at 1 µg/ml. The detection was performed using the secondary antibody donkey anti-mouse Alexa 647 (Thermo Fisher Scientific, Cat# A-31571). All steps, from immunolabeling up to imaging and quantification were performed blind.

Images were acquired on an inverted Leica DMi8 microscope (Leica Microsystems) equipped with a Flash 4.0 sCMOS camera (Hamamatsu, Hamamatsu, Japan). The illumination system used was a Cool LED PE-4000 (CoolLED, Andover). The objective used was a HC PL APO CS2 ×63 oil 1.4 NA DIC.

Images quantifications were performed using tasks automatization with MetaMorph. Following a background subtraction, the images were automatically thresholded in order to determine several ROIs around the neurons. In these regions, we applied an automatic threshold to determine positive objects for the eGFP (Xph constructs expression) and Cy5 (PSD-95 immunolabeling) channels. We measured the average intensity of these objects. We also measured the integrated fluorescence intensity of the entire regions.

## Correlation

After nucleofection at 0 DIV of 1.5 µg of Xph15- or Xph20-eGFP with 1.5 µg of PSD-95-mScarlet-I, E18 rat Banker neuronal cultures were fixed using 4% PFA at 16 DIV.

Images were acquired on an inverted Leica DMi8 microscope (Leica Microsystems) equipped with a Flash 4.0 sCMOS camera (Hamamatsu). The illumination system used was a Cool LED PE-4000 (CoolLED). The objective used was a HC PL APO CS2 ×63 oil 1.4 NA DIC.

The quantification of the images was performed using tasks automatization with MetaMorph. Following a background subtraction, the images were automatically thresholded in order to determine several ROIs around the neurons. Then inside these regions, the intensity of the positive objects for the recombinant binders (eGFP for Xph15, Xph20) and PSD-95 (mScarlet-I) was measured and compared in both green and red channels.

## STED

Fixed E18 rat neuronal cultures (DIV21) expressing GFP-tagged Xph20 were imaged with a glycerol immersion objective (Plan Apo 93× NA 1.3 motCORR). Cells were immunolabeled with MAP2

(Synaptic Systems 188006 and anti-chicken AF594, Thermo Fisher A11042) to identify dendritic draft. A 660 nm wavelength laser was used for GFP depletion. Acquisition parameters were 20 nm pixel size, four times accumulated average per line and 200 Hz scan speed.

E18 rat Banker cultures expressing mNeonGreen-tagged Xph15 or Xph20 were imaged at 37°C in Tyrode's solution. Live staining of rat hippocampal neurons transfected with both cytosolic eGFP and Xph15-SNAP-tag at DIV 10 was adapted from *Bottanelli et al., 2016*. Transfected neurons seeded on 18 mm coverslips at DIV 17 were incubated at 37°C in the presence of 2 µl of aliquoted stock solution of the fluorescent ligand BG-SiR diluted in 250 µl of conditioned Neurobasal medium (final 5 µM BG-SiR). After 1 hr incubation, neurons were washed three times with 1 ml of $CO_2$-equilibrated Neurobasal medium. Each wash was corresponding to a minimal 15 min incubation time with the fresh medium to ensure removing all excess of unbound fluorescent ligand. Coverslip with neurons was then mounted in a Ludin chamber filled with 600 µl of pre-warmed Tyrode medium.

Live neurons were imaged with an inverted Leica SP8 STED microscope equipped with an oil immersion objective (Plan Apo 100× NA 1.4), white light laser 2 (WLL2, 470–670 nm, 80 MHz frequency, ca. 200 ps pulse duration), and internal hybrid detectors. A 775 nm pulsed wavelength laser (80 MHz frequency, ca. 600 ps pulse duration) was used to deplete SiR dye excited by the 647 nm laser line. To preserve neuron health, low STED power was used: time-averaged measurements of STED laser power at the focal plane were showing a value lower than 20 mW (using S120C probe from Thorlabs). Other acquisition parameters were 19 nm pixel size; 16 times average per line; bidirectional 400 Hz scan speed. Final images were processed in ImageJ as follows: gentle convolution using convolve plugin with a 3 × 3 kernel (1 1 1, 1 10 1, 1 1 1), slight chromatic correction to align GFP image with STED capture. Gamma correction of 0.5 was applied on neuron large view image to help seeing small synapses stained with SiR.

## (spt)PALM

Live or fixed (PFA 4%) cells were mounted in a Ludin chamber filled with 1 ml of Tyrode's solution (in mM): 10 HEPES, 5 KCl, 100 NaCl, 2 $MgCl_2$, 2 $CaCl_2$, 15 glucose (pH 7.4), and imaged at 37°C. An inverted microscope (DMi8, Leica, Germany) equipped with a TIRF objective (160 × 1.43 NA, Leica), a Ilas² TIRF device, and an Evolve EMCCD camera (Roper Scientific, Evry, France) was used for (spt)PALM recordings. Neurons expressing mEos3.2-tagged constructs were photo-activated using a 405 nm laser, and the resulting photo-converted single-molecule fluorescence signal was excited with a 561 nm laser. The power of the 405 nm laser was adjusted to keep the number of the stochastically activated molecules constant and well separated during the acquisition. Images were acquired by image streaming for up to 4000 frames (sptPALM) or up to 20,000 frames (PALM) at a frame rate of 50 Hz using MetaMorph software (Molecular Devices), and analysis was performed with a homemade software developed under MetaMorph and kindly provided by J.B. Sibarita (Interdisciplinary Institute for Neuroscience).

## SMLM analysis

Localization and tracking reconnection of ATTO-647N (uPAINT) or mEos3.2 (PALM) signals were performed using homemade software developed as a MetaMorph plugin and kindly provided by J.B. Sibarita (Interdisciplinary Institute for Neuroscience) (*Kechkar et al., 2013*). Single-molecule fluorescence could be identified by occurrence of fluorescence in the red channel and the defined minimum duration of fluorescence. Trajectories were reconstructed using a simulated annealing algorithm (*Racine et al., 2006*), taking into account molecule localization and total intensity. Diffusion coefficients were calculated by linear fit of the first four points of the mean square displacement plots.

PALM clusters analysis was performed using SR-Tesseler software as described previously .

## DNA-PAINT

E18 rat hippocampal neurons transfected with Xph20-SNAP and cytosolic eGFP were fixed at DIV 14–16 with 4% PFA in PBS for 20 min. Neurons were then quenched with 150 mM glycine for 20 min, followed by simultaneous blocking and permeabilization for 90 min in PBS supplemented with 0.2% Triton-X-100 and 3% BSA. For SNAP labeling, cells were incubated with 1 µM of SNAP-ligand-modified DNA oligomer in PBS supplemented with 0.5% BSA and 1 mM DTT for 1 hr.

Neurons were imaged at 25°C in a Ludin chamber with an inverted motorized microscope (Nikon Ti) equipped with a CFI Apo TIRF 100× oil, NA 1.49 objective and a perfect focus system PFS-2, allowing long acquisition in TIRF illumination mode. For DNA-PAINT nanoscopy, neurons expressing Xph20-SNAP were first incubated for 15 min with 90 nm Gold Nanoparticles (Cytodiagnostics) to serve as fiducial markers. Xph20-SNAP was then visualized with Cy3b-labeled DNA imager strands, added to the Ludin chamber at variable concentrations (2–5 nM), as described previously (*Schnitzbauer et al., 2017*). Cy3B-labeled strands were visualized with a 561 nm laser (Cobolt Jive). Fluorescence was collected by the combination of a dichroic and emission filters (dichroic: Di01-R561; emission: FF01-617/73, Semrock) and a sensitive sCMOS (scientific CMOS, ORCA-Flash4.0, Hammatasu). The acquisition was steered using MetaMorph software (Molecular Devices) in streaming mode at 6.7 Hz. GFP was imaged using a conventional GFP filter cube (excitation: FF01-472/30; dichroic: FF-495Di02; emission: FF02-520/28, Semrock). Super-resolution DNA-PAINT reconstruction and drift correction were carried out as described before using the software package Picasso (*Schnitzbauer et al., 2017*).

## Calcium signaling imaging

Imaging of GCaMP6f and Xph-GCaMP6f was carried out in E18 rat hippocampal-dissociated culture nucleofected with the appropriate DNA on the day of the culture. Neurons were imaged at 13–18 DIV using a Nikon inverted microscope (Ti Eclipse) with an EMCCD camera (Evolve 512, Photometrics) controlled using MetaMorph software (Molecular Devices) and equipped with a ×60/1.49 NA oil-immersion objective (Nikon). Images were acquired at a rate of ~50 Hz. The imaging chamber (Ludin Chamber, Life Imaging Services) was perfused with extracellular buffer containing (in mM) 130 NaCl, 2.5 KCl, 3 $CaCl_2$, 0.1 $MgCl_2$, 10 glucose, 10 HEPES, 0.001 TTX, 0.05 PTX (pH adjusted to 7.4 with NaOH and osmolarity adjusted to 280 mOsm) at RT. The fluorophores were excited with 488 nm laser lines and imaged with the appropriate filters.

E18 rat hippocampal neurons were electroporated with GCaMP7f or Xph20-GCaMP7f and Homer1c-Dsred using the 4D Nucleofection system (Lonza) at DIV 0, seeded on 18 mm glass cover-slips, and cultured for 15 d. Imaging was performed by placing coverslips in a Ludin observation chamber (Life Imaging Services) in $Mg^{2+}$-free Tyrode's solution (15 mM D-glucose, 108 mM NaCl, 5 mM KCl, 2 mM $CaCl_2$, and 25 mM HEPES, pH 7.4) containing 20 µM glycine inside a thermostatic chamber (37°C) placed on an inverted microscope (Nikon Ti-E Eclipse) equipped with an EMCCD camera (Evolve 512, Photometrics) controlled using MetaMorph software (Molecular Devices) and equipped with a ×60/1.49 NA oil-immersion objective (Nikon). Fluorescence was collected using a mercury lamp (Nikon Xcite) and appropriate filter sets (SemROCK).

For *Figure 8d*. quantification of synaptic enrichment was performed by segmenting Homer1c-DsRed clusters (synapses). These regions were transferred onto the GCaMP7f signal, and the average intensity of GCaMP7f in these synaptic regions was measured and divided by the average intensity of the shaft area containing no homer-positive signal.

## Acknowledgements

This research was financially supported by grants from the Centre National de la Recherche Scientifique, the Conseil Régional d'Aquitaine, the France BioImaging national infrastructure (grant ANR-10-INBS-04), the Agence Nationale de la Recherche (CheMoPPI, ANR-13-BS07-0019-01, OptoXL, ANR-16-CE16-0026) to CP and MS, the European Research Council (Dyn-Syn-Mem 787340) to DC, the Labex BRAIN (ANR-10-LABX-43) to CR, and Fondation pour la Recherche Médicale fellowship to BC. We also thank the IINS cell culture facility and Emeline Verdier for technical assistance, the Biochemistry and Biophysics Core Facility of the Bordeaux Neurocampus both funded by the Labex BRAIN (ANR-10-LABX-43), the Charite Viral Core Facility for AAV production and Y Ruffin for technical assistance. Financial support from the IR-RMN-THC Fr3050 CNRS for conducting the research is gratefully acknowledged. We are grateful to M Goillandeau for the upgrade of the mini analysis software, Dolors Grillo-Bosch for peptide synthesis, Kashyap Maruthi for making protein samples for the NMR studies, and Fabrice Cordelières for the FRAP analysis macro development.

## Additional information

### Funding

| Funder | Grant reference number | Author |
|---|---|---|
| Agence Nationale de la Recherche | ANR-13-BS07-0019 | Cameron D Mackereth |
| Agence Nationale de la Recherche | ANR-16-CE16-0026 | Matthieu Sainlos |
| European Research Council | Dyn-Syn-Mem 787340 | Daniel Choquet |
| Agence Nationale de la Recherche | ANR-19-CE11-0025 | Ingrid Chamma |
| Fondation pour la Recherche Médicale | fellowship | Benjamin Compans |

The funders had no role in study design, data collection and interpretation, or the decision to submit the work for publication.

### Author contributions

Charlotte Rimbault, Formal analysis, Investigation, Methodology, Writing – original draft, Writing – review and editing, C.R. performed the biophysical experiments and generated the constructs, with the help of C.B., C.G., I.G; Christelle Breillat, Investigation, Methodology, Writing – review and editing, C.B. conducted the initial imaging experiments and performed the immunostaining, knock-down and correlation experiments together with the interactomics sample preparation; Benjamin Compans, Formal analysis, Investigation, Methodology, Writing – review and editing, B.C. performed the uPAINT and PALM/sptPALM experiments; Estelle Toulmé, Formal analysis, Investigation, Methodology, Writing – review and editing, E.T. performed the electrophysiology experiments. E.T. and I.C. performed the GCaMP experiments; Filipe Nunes Vicente, Investigation, Methodology, Writing – review and editing, J.F.N.V. performed the DNA-PAINT experiments with the supervision of G.G; Monica Fernandez-Monreal, Investigation, Methodology, Writing – review and editing, M.F.M. performed the FRAP experiments; Patrice Mascalchi, Investigation, Methodology, Writing – review and editing; Camille Genuer, Virginia Puente-Muñoz, Isabel Gauthereau, Eric Hosy, Investigation; Stéphane Claverol, Formal analysis, Investigation, Methodology, Writing – review and editing, S.C. performed the proteomics analysis; Gregory Giannone, Supervision, Investigation, Methodology, Writing – review and editing; Ingrid Chamma, Formal analysis, Investigation, Methodology, Writing – review and editing; Cameron D Mackereth, Formal analysis, Funding acquisition, Investigation, Methodology, Writing – review and editing, C.D.M. designed NMR experiments, performed all NMR experiments and analyzed data; Christel Poujol, Funding acquisition, Investigation, Methodology, Writing – review and editing, C.P., M.F.M. and P.M. performed the STED experiments. C.B., V.P. and C.P. performed the FRET/FLIM experiments; Daniel Choquet, Matthieu Sainlos, Conceptualization, Formal analysis, Supervision, Funding acquisition, Validation, Methodology, Writing – original draft, Project administration, Writing – review and editing

### Author ORCIDs

Charlotte Rimbault ⬚ https://orcid.org/0000-0002-4760-8430
Benjamin Compans ⬚ https://orcid.org/0000-0001-7823-1499
Monica Fernandez-Monreal ⬚ https://orcid.org/0000-0001-7278-448X
Eric Hosy ⬚ http://orcid.org/0000-0002-2479-5915
Daniel Choquet ⬚ http://orcid.org/0000-0003-4726-9763
Matthieu Sainlos ⬚ https://orcid.org/0000-0001-5465-5641

### Ethics

All experiments were performed in accordance with the guidelines established by the European Communities Council (Directive 2010/63/EU of September 22, 2010) and were approved by the Animal Experimental Committee of Bordeaux (CE50).

**Decision letter and Author response**
Decision letter https://doi.org/10.7554/eLife.69620.sa1
Author response https://doi.org/10.7554/eLife.69620.sa2

## Additional files

### Supplementary files
- Supplementary file 1. FRAP data.
- Supplementary file 2. List of plasmids used in this work (c = commercial source; *from this study). 1. *Rimbault et al., 2019*. 2. *Sainlos et al., 2011* 3. *Boyden et al., 2005* 4. *Gross et al., 2013* 5. *Chen et al., 2013* 6. *Dana et al., 2019* 7. *Mondin et al., 2011*.
- Supplementary file 3. Primers used in this study.
- Transparent reporting form

### Data availability
All data generated or analysed during this study are included in the manuscript and supporting files.

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
