## [Editor Report]

This is a valuable manuscript that develops binders for visualizing postsynaptic protein PSD95 endogenously using a variety of microscopy approaches. Compelling evidence is provided for validating the use of these new imaging probes. These probes should prove useful for visualizing the post-synaptic density in both live and fixed neuronal cells using live cell imaging as well as super-resolution microscopy.

---

## [Decision Letter]

**Decision letter after peer review:**

Thank you for submitting your article "Engineering paralog-specific PSD-95 synthetic binders as potent and minimally invasive imaging probes" for consideration by *eLife*. Your article has been reviewed by 3 peer reviewers, and the evaluation has been overseen by a Reviewing Editor and Lu Chen as the Senior Editor. The following individual involved in review of your submission has agreed to reveal their identity: Helge Ewers (Reviewer #1).

Essential revisions:

All the reviewers agreed that the probes described here were of general interest to the field of neuroscience and warrant publication in *eLife* as long as certain aspects of the work can further be strengthened as outlined below and in the reviewer comments. Please provide a revised manuscript with a point by point response to the reviewer comments, paying particular attention to the points below:

1) Reviewers felt that the specificity of the binders in neurons should be better demonstrated. Some ways to validate specificity would be to compare the binders with existing probes like antibodies and compute co-localization ratios as well as to use shRNA knock down approaches.

2) In line with the point above, reviewers felt that the labeling stoichiometry should be better characterized and described. Expression levels seem to impact the labeling stoichiometry either leading to non-specific filling of neurons or sub-stoichiometric labeling. Labeling efficiency/specificity should be characterized as a function of the expression level and more thoroughly described in the manuscript.

3) While reviewers agreed that demonstration of novel biological applications would be beyond the scope of this manuscript, a more detailed discussion of how these probes would enable novel biological applications that cannot be achieved with existing probes should be included in the manuscript.

*Reviewer #1 (Recommendations for the authors):*

This is important and well executed work that provides an excellent new tool to study an important synaptic molecule and at the same time highly specific delivery of sensors with minimal perturbance. Hardly ever are biological tools characterised this thoroughly. Very valuable and important work.

The reviewer has but few comments:

One page 12 something seems to have gone wrong in formatting, the first paragraph hangs in the air and does not end.

There are a few typos:

– the bivalent psd-95 binder is certainly not Fluorescein-derived, but rather derivatized with Fluorescein or similar.

– Figure 2 Supplementary Figure 1 is named Figure 1- supplementary Figure 1.

*Reviewer #2 (Recommendations for the authors):*

It was a great pleasure to follow the systematic, thorough and careful biophysical and functional characterization of the three unique and specific PSD95 monobody-binders and their successful high-end microscopy application combined with five different reporting systems. The experimental design, data analysis and the description and visualization of the results are on the highest level.

In my personal view widely used probes such as SIR-Actin, Phalloidin or DAPI/Höchst demonstrate that functional interference and labeling can be very well separated or tolerated as long as the probes are simple to use and allow for high contrast and ideally close to stoichiometric labeling. At the same time I fully appreciate the focus on possible functional interference and compatibility with high-end microscopy.

In summary I fully support immediate publication in *eLife* after addressing or clarifying the following major points:

1) The title promises “potent” and “specific” binders and in the abstract and throughout the manuscript the authors claim to provide “efficient” binders with “strong colocalization” and accordingly the authors conclude “…that the three monobodies label PSD-95 efficiently”.

Here I was missing the necessary determination of Pearsons and/or Manders coefficients for the introduced probes compared to a genetic stoichiometric label or conventional antibodies. The data and analyses presented in 3a to 3d only determine “% of common pixels within a probe labelled object with PSD95 immostaining” – thus only half of the necessary data/analysis. This appears important as the images in 3a suggest that monobodies, when expressed under regulated conditions, indeed only achieve a fractional binding of the endogenous target. In addition, unregulated overexpression appears to result in “filling the neuron”.

A complete data analysis should include the information how well the “robe co-localized with the target and how well the target co-localized with the probe ideally for both expression systems.

2) The authors emphasize that currently available and broadly applied PSD95 targeting affinity proteins (Gross 2013, Fukata 2013) are less well characterized in terms of epitopes and binding mechanism. Since the existing probes have been applied broadly and with great success it would have been helpful to the community to resolve the possible issue of the authors regarding a possible functional interference by including any of the existing monobodies in selected experiments.

Here it would have been helpful to learn based on what evidences the authors speculate existing antibodies and/or monobodies do display functional interference and specifically interference with receptor binding?

Using a set of high-end biophysical technologies in a whole array of experiments the authors exclude interference of monobody binding and receptor binding of PSD95. Yet, in its role as central scaffold of the excitatory synapse PSD95 engages in many more critical protein-protein interactions than only the receptors. To this end, I was wondering why the authors did not demonstrate the absence of functional interference and specifically protein-protein-interactions by determining the interactomes of PSD95 in presence and absence of the monobodies in the same way as in the original report and characterization of the monobodies by the same authors in Nat Commun 2019. Along the same lines, the earlier published MS/MS data of monobodies Xph0- versus Xph20- pulldowns do actually suggest the interference of at least one of the monobodies with distinct protein-protein-interactions (PPIs) but not the receptor binding.

3) The authors evaluate the use of the monobodes in STED (Figure 4 and 6) and PALM (Figure 5). For all these experiments I was missing the necessary comparison with conventional PSD95 probes or a stoichiometric fluorophore tag quantifying to what extend the detections actually correlate to endogenous PSD95. If these experiments were not conducted some of the claims should at least be be tuned down e.g. "close stoichiometric labeling of the endogenous protein” “accurate snapshot of the PSD95 nanoscale distribution” “we are essentially detecting probes bound to PSD95” e.g. by adding “proposing” “suggesting” “indicating” “at least fractional labeling”

4) The authors claim that they were “… taking advantage of (…) their unique paralog-specific recognition properties…” This appears to be somewhat contradicting their earlier statements (Nat Commun 2019 from the same authors) eg: “If we compare the 10FN3 domain part of our modulators to some of the commercially available PSD-95-specific anibodies, epitopes are also found in fragments that encompass the same surface region targeted by Xph15, Xph18 and Xph20, suggesting shared epitopes that may rely on the same residues to achieve specificity”. Along the same lines it appears counter-intuitive to expect that the reported monobodies are not interfering with any endogenous PPI. Indeed this was also highlighted by the authors in the original report of the monobodies: “Our results suggest that parts of the first PDZ domain of PSD-95 constitute hot-spots for PPI as they appear to be involved in the binding of all the isolated clones.” It will be helpful for the readers of the first report to have this clarified.

5) Due to the existence of widely applied intrabody-based approaches for the chosen target one might assume the provided probes provide only a limited improvement over the current state-of-the-art. Indeed, the new probes were not applied in this study to answer biological questions that could not be answered with the existing probes. Instead the authors pioneer the application of the probes in high-end imaging. Yet, since the author do speculate about future applications e.g. in their last sentence they my consider do either explain to the non-expert the tremendous challenge of developing a non-interfering probe with specificity in cellular-context or alternatively add e.g. one sentence how future applications such multiplexing imaging strategies will benefit and how they will improve our neurological understanding.

*Reviewer #3 (Recommendations for the authors):*

– The specificity of the binders for endogenous PSD-95 should be directly assessed. The authors, state “We interpret this difference as a direct benefit from the specificity of the Xph clones for PSD-95 while PSD95.FingR, which can also bind SAP97 and SAP102, may also report to some extent the presence of these two paralog proteins”. This would indeed be an advantage of the presented binders over PSD95.FingR, but remains unaddressed here. In fact, some images (in particular Figure 3-supp1 panel a) show that the binders also label axons (even with the transcription regulation of the probes) even though PSD-95 has not been described to be localized in axons. How often does this occur? In general, the specificity of these probes for labeling endogenous PSD-95 should be made more convincing, for instance by depleting endogenous PSD-95.

– All the presented applications merely confirm the extensive existing literature on PSD-95. Could the authors demonstrate an advantage of this probe by showing a conceptual advancement that was not possible before?

– The use of the probe for anchoring GcaMP at the synapse seems beneficial, but how does this compare to using soluble GcaMP? No analysis of calcium traces is shown to demonstrate the advantage of this approach.

– The electrophysiology data presented in Figure 2 requires further analysis for it to be fully convincing. Representative traces appear to show differences in amplitude as well as differences in filtering. Additionally, traces for all binders need to be shown. From the cumulative distribution in Figure 2b, it seems that Xph18 leads to an overall increase in mEPSC amplitude. This needs to be tested using an appropriate statistical test on the cumulative distribution to clarify whether the Xph18 values are significantly different. Furthermore, the mean amplitude for Xph20 reveals a tendency to decrease. Since the variation is relatively large, it would be valuable to increase the number of analyzed neurons to confirm this point. Finally, as Xph20 is used later in the manuscript and the data from the recorded neurons is already included in Figure 2, the data on frequency and rise/decay time for Xph20 should be included as well.

– In Figure 2e, the distribution of diffusion coefficients shows an increase in mobile tracks for the binders but this is not apparent from the quantification in f. Further analysis and statistics on the mobile tracks should be included to clarify this point.

– In Figure 3, Xph18 labeling is significantly dimmer and labels smaller puncta than the other probes. Quantification of the size of the puncta is required to assess whether this is due to variability or differences in binding/labeling of the probes. In Figure 3b, it is unclear how these values compare to the enrichment of endogenous PSD-95 labeling. Additionally, the quantification of co-localization shown in panels c and d is not clearly described in the methods, and therefore hard to justify. This needs to be clarified, also because the values do not appear to correlate with what is shown in the representative images: the co-localization of PSD95.FingR with PSD-95 labeling seems comparable to what is shown for the binders. The number of included neurons also seem to vary extensively, this should be more equal among the groups to correctly compare the distribution of values in the quantifications.

– The FRAP experiment in figure 3e, f is based on a very low n of spines, this should be repeated in at least a number of neurons from 2 or more individual cultures. The FRAP experiment reveals a large pool of unbound Xph-derived probes, which can confound analysis of PSD-95. Particularly here it is important to know if this is in the context of the transcriptional repression system. The 80% recovery observed in the FRAP experiments is also contradictory with the 20% mobile pool of Xph-labeled molecules shown in figure 5 PALM experiments. This should be clarified.

– The STED data in Figure 4f and g show very divergent synaptic structures from other panels showing STED- and PALM-resolved PSDs. Additionally, the zoom-ins in Figure 6 show structures strikingly different from other PSDs throughout the manuscript. These points should be addressed and clarified whether they arise from the probes themselves or the techniques used.

---

## [Author Response]

Essential revisions:1) Reviewers felt that the specificity of the binders in neurons should be better demonstrated. Some ways to validate specificity would be to compare the binders with existing probes like antibodies and compute co-localization ratios as well as to use shRNA knock down approaches.

To address this point, we have performed two new sets of experiments, one where we determine the fluorescence correlation between the probes and a PSD-95 FP-fusion and another where we used a PSD-95 shRNA KD approach (new Figure 4). We have also improved the immunostaining experiments (Figure 3a-d). The knock-down approach, in complement to the initial in vitro experiments, clearly demonstrate that the probes are specific for PSD-95.

2) In line with the point above, reviewers felt that the labeling stoichiometry should be better characterized and described. Expression levels seem to impact the labeling stoichiometry either leading to non-specific filling of neurons or sub-stoichiometric labeling. Labeling efficiency/specificity should be characterized as a function of the expression level and more thoroughly described in the manuscript.

In order to better characterize and describe the effective labeling stoichiometry, we rely on an ensemble of experiments: the PSD-95 shRNA KD approach (new Figure 4) indicates that the signal is lost in absence of PSD-95, the immunostaining (Figure 3 a-d) and PSD-95 FP fusion (new Figure 4) are consistent with a strong correlation of the labeling, spt-PALM experiments (Figure 6f-j) suggest that most probes are bound and PALM experiments (Figure 6a-e) provide a quantitative estimation of PSD-95 synaptic copy number consistent with other techniques. We note that the variability in expression levels is integrated in the correlation experiments (Figure 4).

3) While reviewers agreed that demonstration of novel biological applications would be beyond the scope of this manuscript, a more detailed discussion of how these probes would enable novel biological applications that cannot be achieved with existing probes should be included in the manuscript.

We have added a paragraph in the Discussion section to specify which type of applications can now be considered with the probes.

Reviewer #1 (Recommendations for the authors):This is important and well executed work that provides an excellent new tool to study an important synaptic molecule and at the same time highly specific delivery of sensors with minimal perturbance. Hardly ever are biological tools characterised this thoroughly. Very valuable and important work.The reviewer has but few comments:One page 12 something seems to have gone wrong in formatting, the first paragraph hangs in the air and does not end.

The end of the paragraph has been reinserted in place.

There are a few typos:– the bivalent psd-95 binder is certainly not Fluorescein-derived, but rather derivatized with Fluorescein or similar.

Corrected in SI Figure 2

– Figure 2 Supplementary Figure 1 is named Figure 1- supplementary Figure 1.

The figures referencing will be updated according to the editors recommendation in due time.

Reviewer #2 (Recommendations for the authors):It was a great pleasure to follow the systematic, thorough and careful biophysical and functional characterization of the three unique and specific PSD95 monobody-binders and their successful high-end microscopy application combined with five different reporting systems. The experimental design, data analysis and the description and visualization of the results are on the highest level.In my personal view widely used probes such as SIR-Actin, Phalloidin or DAPI/Höchst demonstrate that functional interference and labeling can be very well separated or tolerated as long as the probes are simple to use and allow for high contrast and ideally close to stoichiometric labeling. At the same time I fully appreciate the focus on possible functional interference and compatibility with high-end microscopy.In summary I fully support immediate publication in eLife after addressing or clarifying the following major points:1) The title promises "potent" and "specific" binders and in the abstract and throughout the manuscript the authors claim to provide "efficient" binders with "strong colocalization" and accordingly the authors conclude "…that the three monobodies label PSD-95 efficiently".

The title has been changed.

Here I was missing the necessary determination of Pearsons and/or Manders coefficients for the introduced probes compared to a genetic stoichiometric label or conventional antibodies. The data and analyses presented in 3a to 3d only determine "% of common pixels within a probe labeled object with PSD95 immostaining" – thus only half of the necessary data/analysis. This appears important as the images in 3a suggest that monobodies, when expressed under regulated conditions, indeed only achieve a fractional binding of the endogenous target. In addition, unregulated overexpression appears to result in "filling the neuron".A complete data analysis should include the information how well the probe co-localized with the target and how well the target co-localized with the probe ideally for both expression systems.

This general point has been addressed with a new set of experiments presented in the new figure 4d-e by analyzing the correlation of the Xph15 and Xph20 probes eGFP signal with the one of a PSD-95-mScarlet-I fusion. In addition, the immunostaining experiments (Figure 3A-d) have been repeated to increase and homogenize the number of data points.

2) The authors emphasize that currently available and broadly applied PSD95 targeting affinity proteins (Gross 2013, Fukata 2013) are less well characterized in terms of epitopes and binding mechanism. Since the existing probes have been applied broadly and with great success it would have been helpful to the community to resolve the possible issue of the authors regarding a possible functional interference by including any of the existing monobodies in selected experiments.Here it would have been helpful to learn based on what evidences the authors speculate existing antibodies and/or monobodies do display functional interference and specifically interference with receptor binding?Using a set of high-end biophysical technologies in a whole array of experiments the authors exclude interference of monobody binding and receptor binding of PSD95. Yet, in its role as central scaffold of the excitatory synapse PSD95 engages in many more critical protein-protein interactions than only the receptors. To this end, I was wondering why the authors did not demonstrate the absence of functional interference and specifically protein-protein-interactions by determining the interactomes of PSD95 in presence and absence of the monobodies in the same way as in the original report and characterization of the monobodies by the same authors in Nat Commun 2019. Along the same lines, the earlier published MS/MS data of monobodies Xph0- versus Xph20- pulldowns do actually suggest the interference of at least one of the monobodies with distinct protein-protein-interactions (PPIs) but not the receptor binding.

We have focused our work on characterizing the monobodies that we have developed and have not conducted experiments to further characterize the one developed by others. To address the point raised by reviewer 2, we have performed an interactomic analysis of PSD-95 in presence and absence of Xph20 (best binding clone). The results are presented in Figure 2i.

We note that the earlier proteomic experiments (Nat Commun 2019) were conducted with engineered monobodies that were fused to PDZ domain-binding motifs and for which we therefore expected interference with their capacity to bind their native partners.

3) The authors evaluate the use of the monobodes in STED (Figure 4 and 6) and PALM (Figure 5). For all these experiments I was missing the necessary comparison with conventional PSD95 probes or a stoichiometric fluorophore tag quantifying to what extend the detections actually correlate to endogenous PSD95. If these experiments were not conducted some of the claims should at least be be tuned down e.g. "close stoichiometric labeling of the endogenous protein" "accurate snapshot of the PSD95 nanoscale distribution" "we are essentially detecting probes bound to PSD95" e.g. by adding "proposing" "suggesting" "indicating" "at least fractional labeling"

We have not systematically reevaluated every probe variants that would just consist in changing the FP (eGFP, mEos3.2, …) and we therefore base the efficiency of the systems used for STED and PALM on the characterizations performed in prior sections with the eGFP versions. These characterizations have been improved with new experiments in the corrected manuscript version.

We note that the three examples ("close stoichiometric labeling of the endogenous protein" "accurate snapshot of the PSD95 nanoscale distribution" "we are essentially detecting probes bound to PSD95") are always done in the manuscript with a clear reference to published results obtained by other techniques or tools that we use as a comparison to our results.

4) The authors claim that they were "… taking advantage of (…) their unique paralog-specific recognition properties…" This appears to be somewhat contradicting their earlier statements (Nat Commun 2019 from the same authors) eg: "If we compare the 10FN3 domain part of our modulators to some of the commercially available PSD-95-specific anibodies, epitopes are also found in fragments that encompass the same surface region targeted by Xph15, Xph18 and Xph20, suggesting shared epitopes that may rely on the same residues to achieve specificity". Along the same lines it appears counter-intuitive to expect that the reported monobodies are not interfering with any endogenous PPI. Indeed this was also highlighted by the authors in the original report of the monobodies: "Our results suggest that parts of the first PDZ domain of PSD-95 constitute hot-spots for PPI as they appear to be involved in the binding of all the isolated clones." It will be helpful for the readers of the first report to have this clarified.

The “unique” properties refer to the fact that they are the only described recombinant domains that can discriminate paralogs. For the interference issue, we have added the interactomics experiments (Figure 2) to extend our analysis capacity beyond chosen canonical binders of PSD-95.

5) Due to the existence of widely applied intrabody-based approaches for the chosen target one might assume the provided probes provide only a limited improvement over the current state-of-the-art. Indeed, the new probes were not applied in this study to answer biological questions that could not be answered with the existing probes. Instead the authors pioneer the application of the probes in high-end imaging. Yet, since the author do speculate about future applications e.g. in their last sentence they my consider do either explain to the non-expert the tremendous challenge of developing a non-interfering probe with specificity in cellular-context or alternatively add e.g. one sentence how future applications such multiplexing imaging strategies will benefit and how they will improve our neurological understanding.

We have added a paragraph in the Discussion section (p18) to present applications of our tools that cannot be achieved with other current tools.

Reviewer #3 (Recommendations for the authors):– The specificity of the binders for endogenous PSD-95 should be directly assessed. The authors, state "We interpret this difference as a direct benefit from the specificity of the Xph clones for PSD-95 while PSD95.FingR, which can also bind SAP97 and SAP102, may also report to some extent the presence of these two paralog proteins". This would indeed be an advantage of the presented binders over PSD95.FingR, but remains unaddressed here. In fact, some images (in particular Figure 3-supp1 panel a) show that the binders also label axons (even with the transcription regulation of the probes) even though PSD-95 has not been described to be localized in axons. How often does this occur? In general, the specificity of these probes for labeling endogenous PSD-95 should be made more convincing, for instance by depleting endogenous PSD-95.

We have performed additional experiments using shRNA to knock-down endogenous PSD-95. The results are presented in Figure 4a-c. We have also removed what appeared as “axonal staining” in SI fig7 as this is never observed in healthy neurons.

– All the presented applications merely confirm the extensive existing literature on PSD-95. Could the authors demonstrate an advantage of this probe by showing a conceptual advancement that was not possible before?

We have added a paragraph in the Discussion section (p18) to present applications of our tools that cannot be achieved with other current tools.

– The use of the probe for anchoring GCaMP at the synapse seems beneficial, but how does this compare to using soluble GCaMP? No analysis of calcium traces is shown to demonstrate the advantage of this approach.

In agreement with the editorial recommendation, we have focused our efforts on other aspects.

– The electrophysiology data presented in Figure 2 requires further analysis for it to be fully convincing. Representative traces appear to show differences in amplitude as well as differences in filtering. Additionally, traces for all binders need to be shown. From the cumulative distribution in Figure 2b, it seems that Xph18 leads to an overall increase in mEPSC amplitude. This needs to be tested using an appropriate statistical test on the cumulative distribution to clarify whether the Xph18 values are significantly different. Furthermore, the mean amplitude for Xph20 reveals a tendency to decrease. Since the variation is relatively large, it would be valuable to increase the number of analyzed neurons to confirm this point. Finally, as Xph20 is used later in the manuscript and the data from the recorded neurons is already included in Figure 2, the data on frequency and rise/decay time for Xph20 should be included as well.

We have performed additional experiments in mouse neurons in order to have measurements for all the clones. The results are now presented in Figure 2a-e. The initial data set for Xph15 and Xph18 obtained in rat neurons is presented in SI Figure 4.

– In Figure 2e, the distribution of diffusion coefficients shows an increase in mobile tracks for the binders but this is not apparent from the quantification in f. Further analysis and statistics on the mobile tracks should be included to clarify this point.

We have added the cumulative distribution of instantaneous diffusion coefficients in Figure SI 5 to illustrate the fact that the difference observed for Xph20 corresponds only to slightly different distribution of the mobile populations with no impact on the mobile vs immobile repartition.

– In Figure 3, Xph18 labeling is significantly dimmer and labels smaller puncta than the other probes. Quantification of the size of the puncta is required to assess whether this is due to variability or differences in binding/labeling of the probes. In Figure 3b, it is unclear how these values compare to the enrichment of endogenous PSD-95 labeling. Additionally, the quantification of co-localization shown in panels c and d is not clearly described in the methods, and therefore hard to justify. This needs to be clarified, also because the values do not appear to correlate with what is shown in the representative images: the co-localization of PSD95.FingR with PSD-95 labeling seems comparable to what is shown for the binders. The number of included neurons also seem to vary extensively, this should be more equal among the groups to correctly compare the distribution of values in the quantifications.

We have performed additional experiments and increased the number of objects analyzed and the number of independent experiments for all conditions in Figure 3b-d. In addition, we have also performed a new set of experiments presented in the new figure 4d-e by analyzing the correlation of the Xph15 and Xph20 probes eGFP signal with the one of a PSD-95-mScarlet-I fusion.

– The FRAP experiment in figure 3e, f is based on a very low n of spines, this should be repeated in at least a number of neurons from 2 or more individual cultures. The FRAP experiment reveals a large pool of unbound Xph-derived probes, which can confound analysis of PSD-95. Particularly here it is important to know if this is in the context of the transcriptional repression system. The 80% recovery observed in the FRAP experiments is also contradictory with the 20% mobile pool of Xph-labeled molecules shown in figure 5 PALM experiments. This should be clarified.

We have repeated and corrected the FRAP experiments with higher spine and cell numbers (Figure 3e-g). For the differences in recovery percentage observed in FRAP and the mobile fraction percentage measurement in spt-PALM, we note that the measurements are not performed on the same time scales (minutes for FRAP and ms for spt-PALM) to be compared to probe-domain complex half-life determined with recombinant proteins (seconds). The difference in time-scale explains that we are not observing the same situation: FRAP and long time-scales reveal the probe capacity to exchange whereas ms time-scale (coupled to a spt-PALM detection bias towards mildly diffusive objects) reflects the probe affinity for its target and stability of the complex in this relatively short time interval. The various time-scales have been clarified in the corrected manuscript every time such notions where critical for the understanding of the results.

– The STED data in Figure 4f and g show very divergent synaptic structures from other panels showing STED- and PALM-resolved PSDs. Additionally, the zoom-ins in Figure 6 show structures strikingly different from other PSDs throughout the manuscript. These points should be addressed and clarified whether they arise from the probes themselves or the techniques used.

Throughout this work, the PSD-95 labeling observed with different microscopy modalities was reproducible and largely independent of the method used despite slight differences associated with technical aspects (signal to noise, acquisition procedure, nature of the dye…). The differences mentioned here are typically associated with the technique used and the choice of representation to illustrate the results. We have added a sentence in the discussion to clarify this point:

p17” Importantly, we note that the PSD-95 labeling observed with the different microscopy modalities here was reproducible and largely independent of the method used despite slight differences associated with technical aspects (signal to noise, acquisition procedure, nature of the dye and its associated effective labeling efficiency…).”